# BP-modified Local Loss for Efficient Training of Deep Neural Networks

**Lianhai Ren**
Department of Mathematics
National University of Singapore
10 Lower Kent Ridge Road, Singapore 119076
`lianhairen@u.nus.edu`

**Qianxiao Li**
Department of Mathematics and
Institute for Functional Intelligent Materials,
National University of Singapore
10 Lower Kent Ridge Road, Singapore 119076
`qianxiao@nus.edu.sg`

## Abstract

The training of large models is memory-constrained, one direction to relieve this is training using local loss, like GIM, LoCo, and Forward-Forward algorithms. However, the local loss methods often face the issue of slow or non-convergence. In this paper, we propose a novel BP-modified local loss method that uses the true Backward Propagation (BP) gradient to modify the local loss gradient to improve the performance of local loss training. We use the stochastic modified equation to analyze our method and show that modified offset decreases the bias between the BP gradient and local loss gradient, but introduces additional variance, which results in a bias-variance balance. Numerical experiments on full-tuning and LoKr tuning on the ResNet-50 model and LoRA tuning on the ViT-b16 model on CIFAR-100 datasets show 20.5% test top-1 accuracy improvement for the Forward-Forward algorithm, 18.6% improvement for LoCo algorithm and achieve only an average 7.7% of test accuracy loss compared to the BP algorithm, with up to 75% memory savings.

## 1 Introduction

Neural networks have seen rapid advancements in recent years, becoming widely applied across various fields. A key algorithm for training deep neural networks is the Backward Propagation (BP) algorithm (Rumelhart et al., 1986), which needs to store the intermediate state in the forward pass to compute the gradient in the backward pass and thus leads to a large memory usage. This issue becomes more severe as the models become deeper and require larger batch sizes, like SimCLR (Chen et al., 2020). Consequently, reducing the memory footprint of training and fine-tuning remains an active area of research (Krizhevsky et al., 2017; Rhu et al., 2016; Malladi et al., 2023).

To mitigate this memory issue, one common approach is gradient accumulation or distributed training strategies, such as Data Parallelism (Zhang et al., 1989; Das et al., 2016) and Pipeline Parallelism (Huang et al., 2019)), which trades memory usage for additional computation time or enlarge the total memory by using multiple devices. An alternative is using local loss (also known as local learning, or layerwise learning) to train the neural networks instead of the full BP (e.g. Greedy InfoMax algorithm (Löwe et al., 2019), Forward-Forward algorithm (Hinton, 2022)), where the backward propagation is applied only to portions of the model, reducing memory consumption. However, this method generally leads to poorer accuracy compared to BP. Recently, the Model Predictive Control (MPC) framework (Ren & Li, 2024) has been proposed to unify the BP algorithm and the local loss method, balancing memory usage and performance through different horizons. It uses the truncated loss to train the neural networks. However, the MPC framework still faces the trade-off between efficiency and performance: smaller horizons require less memory but result in lower accuracy.

Similar to leveraging global information to improve local loss (Lorberbom et al., 2023), we propose a BP-modified local loss method. Our method integrates the true BP gradient to modify the local loss gradient. Using stochastic modified equations (Li et al., 2017; 2018), we show that this modification reduces bias between the gradients but introduces additional variance, thus resulting in a bias-variance

trade-off. Through an analysis based on the Ornstein–Uhlenbeck process and one-step loss, we further illustrate this balance and derive the explicit update equations for the scaling factor $\lambda_h$ and $a_h$ by minimizing the balance. Numerical experiments on full-tuning and LoKr tuning on the ResNet-50 model and LoRA tuning on the ViT-b16 model on the CIFAR-100 dataset show up to 28% improvement in test top-1 accuracy for the Forward-Forward algorithm and 29% improvement for the LoCo algorithm, with only 7.7% average test accuracy loss compared to the BP algorithm while saving 75% memory.

The main contributions of this paper are as follows:

- We introduce a novel local loss training algorithm that significantly improves performance with minimal additional memory overhead.

- We provide a theoretical analysis using the stochastic modified equation, illustrating the bias-variance trade-off and deriving optimal scaling factors.

- Numerical experiments on both CNN and vision transformer models using different tuning methods verify that our method can improve the performance of the local loss methods with minimal additional memory usage.

The rest of the paper is organized as follows. In Section 2, we review the related work. Section 3 introduces the proposed BP-modified local loss method. The theoretical analysis is in Section 4 In Section 6, we present the numerical experiments. Section 7 discusses the limitations and future directions.

## 2 LITERATURE REVIEW

To address the memory issue of the BP algorithm, many local loss methods have been proposed (Belilovsky et al., 2019). For example, Löwe et al. (2019) proposed the Greedy InfoMax (GIM) learning approach which uses local constructive loss to train the model in self-supervised learning. Belilovsky et al. (2019) greedily train each convolution layer at one time and construct the model layerwise. Hinton (2022) proposed the Forward-Forward algorithm using simple local loss to train the MLP model. However, these methods suffer performance issues compared to the BP algorithm since the local loss cannot give the true BP gradient. Recently, Ren & Li (2024) proposed the MPC framework that unifies the BP algorithm and the local loss method. The MPC framework uses the "loss split" method to create local loss from terminal loss and truncated loss. It uses different lengths ("horizon" in the language of MPC) of truncated loss to balance memory usage and performance. It finds the convergence of the gradient to the true gradient in the large horizon. However, the MPC framework still has the efficiency-performance trade-off: small horizons use less memory but result in larger losses.

To improve the performance of the local loss methods, one natural idea is to offer additional information. The LoCo algorithm (Xiong et al., 2020) uses the adjacent two blocks in the model for back-propagation to indirectly pass the global information. Lorberbom et al. (2023) uses the information from other layers to update the threshold in the Forward-Forward algorithm. The proposed BP-modified local loss method uses the difference between the true gradient and local loss gradient in another small batch to modify the original local gradient and reduce its bias thus improving the performance of the original local loss training method.

The implementation of our method uses the delayed update strategy, i.e. the modified term only updates periodically in a larger batch, which is similar to the control variate methods such as SVRG (Johnson & Zhang, 2013), SARAH (Nguyen et al., 2017), and Control variate forward gradient (Arisaka & Li, 2024). Both our methods and theirs include some additional parameters that are expensive in computation (the control variate in their method and the modified term in our method), and an inner loop where these parameters stay unchanged. However, the motivation of the control variate methods is to reduce the variance while our method tends to reduce the bias in the expense of adding variance.

# 3    METHOD: BP-MODIFIED LOCAL LOSS

In this section, we first introduce the deep learning training problem and the MPC framework. Then, we present our BP-modified local loss approach. We consider the "feed-forward" neural network with $T$ layers or blocks, where each block is sequentially connected as follows:

$$x_{t+1} = f_t(x_t, u_t),\tag{1}$$

where $t \in \{0, ..., T-1\}$ denotes the block index, $x_t \in \mathbb{R}^{n_t}$ denotes the input of the $t$-th block, (with $x_0$ being the model input and $x_T$ the model output) , and $u_t \in \mathbb{R}^{m_t}$ are the trainable parameters in the $t$-th block, the function $f_t : \mathbb{R}^{n_t} \times \mathbb{R}^{m_t} \to \mathbb{R}^{n_{t+1}}$ represents the forward mapping of the $t$-th block. Moreover, we denote the combination of head and final loss as $L(x_T)$, omitting possible parameters, targets/labels, and regularization terms for ease of notation. We assume $L$ is compatible with $x_t$ for all $t = 1, \cdots, T$. By compatibility of the loss $L$, we mean it can accept the outputs of all blocks, i.e. $L(x_t)$ is valid for $\forall t = 1, \cdots, T$. For further details of $L$, refer to Appendix A

Recently, Ren & Li (2024) introduced the MPC framework that unifies BP and the local loss method. Since this framework includes many other local losses like Forward-Forward algorithm (Hinton, 2022) and LoCo algorithm (Xiong et al., 2020), we use this framework to present our proposed method, which applies generally to local loss training algorithms.

The MPC framework uses a "loss split" technique to decompose the terminal loss into local losses:

$$l_t(x_t, u_t) = L(f_t(x_t, u_t)) - L(x_t) = L(x_{t+1}) - L(x_t),\tag{2}$$

and then define the truncated loss:

$$L_{h,t}(x_t, u) = \sum_{s=0}^{h-1} l_{t+s}(x_{t+s}, u_{t+s}) = L(x_{h+t}) - L(x_t)\tag{3}$$

where $u = (u_0^\top, \cdots, u_{T-1}^\top)^\top \in \mathbb{R}^m$ represents all the weights in the model, $m = \sum_{t=0}^{T-1} m_t$, and $h \in \{1, \cdots, T\}$ is the hyper-parameter that controls the length of the truncated loss. We assume $x_t = x_T$ when $t > T$ and remove the constraint $t \leq T$ and the minimum argument with $T$ for simplicity. The gradient of $u_t$ under horizon $h$ is defined as the corresponding gradient of $L_{h,t}$:

$$g_{h,t} \triangleq \nabla_{u_t} L_{h,t}(x_t, u) = \nabla_{u_t} L(x_{h+t}).\tag{4}$$

From Eq. (4), we know that when $h = T$, $g_{T,t}$ correspond to the gradient of the terminal loss, reducing to traditional BP, i.e. $g_{T,t} = g_{\mathrm{BP}}$; when $h = 1$, $L_{t,1}(x_t, u) = l_t(x_t, u_t) = L(x_{t+1}) - L(x_t)$, which reduces to the Forward-Forward algorithm (Hinton, 2022). Further, the LoCo algorithm (Xiong et al., 2020) can be seen as the case when $h = 2$. For a detailed example and comparison, please refer to (Ren & Li, 2024).

The MPC framework adjusts the accuracy-memory trade-off through horizon $h$. Since the gradient of the $t$-th block $g_{h,t}$ is $\nabla_{u_t} L(x_{h+t})$, a smaller $h$ uses less global information thus less accurate and a larger horizon $h$ receives more global information and gets better performance. However, the gradient $g_{h,t}$ only depends on the loss $x_{t+h}$, so the gradient only needs to back-prop from $L(x_{t+h})$ to $u_t$ through $h$ blocks, reducing memory usage to $O(h)$, i.e., a smaller horizon will save more memory.

## 3.1    BP-MODIFIED LOCAL LOSS

To retain the memory efficiency while improving the performance of small horizons, we propose a BP-modified local loss. This is especially useful for the $h = 1$ case, i.e. Forward-Forward case, which has minimal memory usage. The method introduces an offset to modify the local loss gradient $g_{h,t}$, as illustrated in Figure 1:

$$\hat{g}_{h,t} = a_{h,t}\mathbb{E}_{\mu_B}[g_{h,t}] + \lambda_{h,t}\mathbb{E}_{\mu_{B'}}[\tilde{g}_{T,t} - a_{h,t}\tilde{g}_{h,t}] \triangleq a_{h,t}\mathbb{E}_{\mu_B}[g_{h,t}] + \lambda_{h,t}\mathbb{E}_{\mu_{B'}}[\Delta\tilde{g}_{h,t}],\tag{5}$$

for all $t = 0, \cdots, T-1$, where $\mu_B$ represents the empirical distribution of the mini-batch with size $B$, i.e. $\mu_B = \frac{1}{B}\sum_{i=1}^{B} \delta_{x_i}$, and $\mu_{B'}$ is the empirical distribution of another independent mini-batch with size $B'$, and letting $g$ denotes the gradient obtained from $\mu_B$, $\tilde{g}$ denotes the gradients obtained from $\mu_{B'}$ and $\hat{g}$ is the modified gradient. $a_{h,t} \in \mathbb{R}$, $\lambda_{h,t} \in [0, 1]$ are two scaling factors, where $a_{h,t}$ scales the original gradient and $\lambda_{h,t}$ scales the offset.

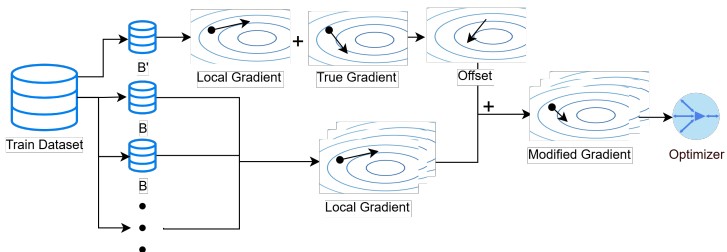

Figure 1: Diagram of the BP-modified local loss method
Usually, there is a large deviation between true gradient $g_T$ and local (loss) gradient $g_h$. The BP-modified local loss method introduces an additional offset $\Delta \tilde{g}_h$ using local gradient and true gradient in batch $B'$ to modify local gradient $g_h$ to modified gradient $\hat{g}_h$ in other batches $B$.

The additional information of the BP-modified local loss method lies in the offset term $\Delta \tilde{g}_{h,t}$, which is the difference between the true gradient $g_{T,t}$ and the local loss gradient $g_{h,t}$ of another mini-batch $\mu_{B'}$. If the difference of gradient in different samples are similar, i.e. $\mathbb{E}_{\mu_B}[g_{h,t} - g_{T,t}] \approx \mathbb{E}_{\mu_{B'}}[g_{h,t} - g_{T,t}]$, then the modified term $\Delta \tilde{g}_{h,t}$ will reduce the bias of local gradient. However, the offset term itself will introduce additional variance. In the following section, we will theoretically analyze the impact of the offset.

Furthermore, to acquire this offset term, we need to sample another mini-batch $\mu_{B'}$ and compute the local gradient $\tilde{g}_{h,t}$ and true gradient $\tilde{g}_{T,t}$ on it. This will increase computation cost and memory usage. Some techniques are used in the algorithm to mitigate the impact on both time and memory usage. For the detailed implementation, please refer to Section 5 and Alg. 1. Storing the offset $\Delta \tilde{g}_{h,t}$ will also need additional memory in the size of trainable parameters, which is negligible especially for CNN models and low-rank tuning cases such as LoRA (Hu et al., 2022).

## 4 THEORETICAL ANALYSIS

In this section, we theoretically analyze the performance of the proposed BP-modified local loss method using the stochastic modified equation of stochastic gradient descent (SGD) (Li et al., 2017; 2018; Mandt et al., 2017) which is applicable for various models and tasks. For clarity, we consider the full set of weights $u$, where the local gradient $g_h$ and true gradient $g_T$ are the concatenation of the block gradient (i.e., $g_h = (g_{h,0}^\top, \cdots, g_{h,T-1}^\top)^\top$). However, the following analysis can apply to each block $t$ individually. Further, we will omit the dependency of gradients $g_*$ and their variance $C_*$ on weights $u$ in the following for simplicity.

Similar to traditional BP, the stochastic modified equation of SGD for horizon $h$ within the original MPC framework can be written as:

$$dU(\tau) = -\mathbb{E}_\mu[g_h]\, d\tau + \sqrt{\frac{\eta}{B} C_h}\, dW_\tau, \tag{6}$$

where $U$ is the first order continuous approximation of weights $u$ as $U(\eta k) \approx u(k)$, $\mu$ denotes the empirical distribution of the whole training data, $\tau$ is time index for the stochastic modified equation, $\eta$ is the learning rate, $B$ is the batch size, and $C_h = Cov(g_h)$[1] is the autocovariance matrix of $g_h$, and $W_\tau$ represents a standard Brownian motion. By Li et al. (2017), we can derive that $U(\eta k)$ is a first-order approximation of $u(k)$ in a weak sense. Please refer to B.1 for details.

To simplify the analysis, we assume that the mini-batch $B'$ for computing offset is independently sampled at each step. The stochastic modified equation for the proposed BP-modified local loss method then becomes:

$$dU(\tau) = -\mathbb{E}_\mu[(1 - \lambda_h)a_h g_h + \lambda_h g_T]\, d\tau + \sqrt{\frac{\eta a_h^2}{B} C_h + \frac{\eta \lambda_h^2}{B'} \tilde{C}_h}\, dW_\tau, \tag{7}$$

---

[1]In the paper, we denote $Cov(X, Y) = \mathbb{E}[(X - \mathbb{E}[X])(Y - \mathbb{E}[Y])^\top]$ denotes the covariance matrix of random vectors $X$ and $Y$, and $Cov(X) = Cov(X, X)$ be the autocovariance matrix of random vector $X$.

where $\tilde{C}_h = Cov(\Delta g_h) = a_h^2 C_h + C_T - a_h C_{h,T}$ is the autocovariance matrix of $\Delta g_h = g_T - a_h g_h$, $C_{h,T} = Cov(g_h, g_T) + Cov(g_T, g_h)$ is the covariance matrix between $g_h$ and $g_T$. The detailed derivation of Eq. (6) and Eq. (7) please refer to Appendix B.1

By comparing Eq. (7) with Eq. (6), we can see that the proposed BP-modified local loss method reduces the bias between the applied gradient with the true gradient from $\|\mathbb{E}_\mu[g_h] - \mathbb{E}_\mu[g_T]\|_2$ to $\|(1 - \lambda_h)\mathbb{E}_\mu[g_h] - \mathbb{E}_\mu[g_T]\|_2$ while introducing additional variance $\frac{\eta \lambda_h^2}{B'}\tilde{C}_h$. This leads to a bias-variance trade-off in the proposed method. We will further illustrate this trade-off using a toy model and then analyze it in the one-step setting in general.

**Remark 4.1** *Note that when $\lambda_h = 0$, Eq. (7) reduces to Eq. (6) since there is no addition offset. When $a_h^t = 0, \lambda_h = 1$, Eq. (7) describes the stochastic modified equation for traditional BP with batch size $B'$ since there is only $g_T$ sampled from $B'$ in Eq. (5).*

## 4.1 TOY EXAMPLE: ORNSTEIN-UHLENBECK PROCESS

To illustrate the bias-variance balance in Eq. (7), we consider an Ornstein-Uhlenbeck (OU) process (Uhlenbeck & Ornstein, 1930). Although it is generally not the true training process of neural networks, this simple model allows for exact calculations which gives important insights into the effect of $\lambda_h$ and bias-variance trade-off that is aligned with the discoveries in the following analysis.

Assuming $u \in \mathbb{R}^n$ and the global loss and local loss with respect to $u$ are:

$$J(u) = \frac{1}{2}u^\top A u \triangleq \frac{1}{2}\|u - u_T^*\|_A^2, \quad J_h(u) = \frac{1}{2}\|u - u_h^*\|_A^2, \tag{8}$$

where $A \in \mathbb{R}^{n \times n}$ is a symmetric positive-definite matrix and the optimal solution are $u_T^*, u_h^*$ respectively, and $u_T^* = \mathbf{0}$. Assuming the gradient variance $V \in \mathbb{R}^{n \times n}$ remains constant throughout the dynamic, the stochastic modified equation of SGD for both global loss $J$ and local loss $J_h$ follows Ornstein-Uhlenbeck (OU) process

$$dU_T(\tau) = -\nabla J(U_T(\tau))d\tau + \sqrt{V}dW_t = -AU_T(\tau)d\tau + \sqrt{V}dW_t. \tag{9}$$

$$dU_h(\tau) = -\nabla J_h(U_h(\tau))d\tau + \sqrt{V}dW_t = -A(U_h(\tau) - u_h^*)d\tau + \sqrt{V}dW_t. \tag{10}$$

Assuming their initial point are the same, i.e. $U_T(0) = U_h(0) = u_0 \in \mathbb{R}^n$.

For the BP-modified version, mimicking Eq. (7) and assuming $a_h = 1$ for simplicity, we add the modified term $Au_h(\tau) - A(u_h(\tau) - u_h^*)$ and the additional variance $\tilde{V}$ (which also remains constant) into Eq. (10), we have:

$$d\hat{U}_h(\tau) = -((1 - \lambda_h)A(\hat{U}_h(\tau) - u_h^*) + \lambda_h A\hat{U}_h(\tau))d\tau + \sqrt{V + \lambda_h^2 \tilde{V}}dW_t \tag{11}$$

Where the drift term is modified towards true gradient (i.e. $Au$), with additional variance and rescaling. The optimal $\lambda_h^*$ and optimal expected loss can be explicitly solved:

$$\lambda_h^*(\mathcal{T}) = \frac{(e^{-A\mathcal{T}}u_0)^\top A u_h^*(\mathcal{T}) + \|u_h^*(\mathcal{T})\|_A^2}{\text{tr}(\tilde{\Sigma}_\mathcal{T}A) + \|u_h^*(\mathcal{T})\|_A^2} = \frac{(e^{-A\mathcal{T}}u_0)^\top A u_h^*(\mathcal{T}) + \text{Bias}}{\text{Var} + \text{Bias}}, \tag{12}$$

$$\min_{\lambda_h}\mathbb{E}[J(\hat{U}_h(\mathcal{T}))] = \mathbb{E}[J(U_h(\mathcal{T}))] - \frac{(\mathbb{E}[U_h(\mathcal{T})]^\top A u_h^*(\mathcal{T}))^2}{2(\text{tr}(\tilde{\Sigma}_\mathcal{T}A) + \|u_h^*(\mathcal{T})\|_A^2)} < \mathbb{E}[J(U_h(\mathcal{T}))], \tag{13}$$

where $u_h^*(\mathcal{T}) = (I - e^{-A\mathcal{T}})u_h^*$, $\tilde{\Sigma}_\mathcal{T} = \lambda_h^2 \int_0^\mathcal{T} e^{-At}\tilde{V}e^{-A^\top t}d\tau$. The derivations of Eq. (12) and Eq. (13) please refer to Appendix B.2. The Bias, Var terms in Eq. (12) just to indicate $\|u_h^*(\mathcal{T})\|_A^2$ is bias-related and $\text{tr}(\tilde{\Sigma}_\mathcal{T}A)$ is variance-related.

For large $\mathcal{T}$, since $e^{-A\mathcal{T}} \to 0$ as $\mathcal{T} \to \infty$, we have $\lambda_h^*(\mathcal{T}) \in (0, 1)$, demonstrating both $\lambda_h = 0$ (original MPC) and $\lambda_h = 1$ (unbiased modification) are suboptimal choices. When bias is dominate, i.e. $\|u_h^*(\mathcal{T})\|_A^2 \ll \text{tr}(\tilde{\Sigma}_\mathcal{T}A)$, $\lambda_h^*$ tends to 1 to reduce the bias, while if the additional variance $\text{tr}(\tilde{\Sigma}_\mathcal{T}A)$ is large compared to the bias $\|u_h^*(\mathcal{T})\|_A^2$, $\lambda_h^*$ tends to 0 to reduce the variance. This is the basic idea of the bias-variance balance which will also appear in the following one-step analysis. Further from Eq. (13), we know that for optimal $\lambda_h^*$, the modified version will have a better-expected loss than the original.

## 4.2 ONE-STEP BIAS-VARIANCE BALANCE

In the previous section, we demonstrate the bias-variance balance inherent in the BP-modified local loss method. However, modeling this through an OU process may oversimplify the dynamics. In this section, we delve deeper into the local bias-variance balance of the proposed method using one-step loss analysis. We postpone the derivation of this section to Appendix B.3 for clarity.

Let $J(u) = \mathbb{E}_\mu[L(x_T)] = \mathbb{E}_\mu[L_{T,0}(x_0, u)]$ be the empirical loss with respect to weights $u$, where $\nabla J(u)$ is assumed to be $\beta-$Lipschitz continuous. Assuming $g(u(k); \xi)$ to be a general gradient estimator whose randomness is controlled by the random variable $\xi$. Using the update rule $u(k+1) = u(k) - \eta g(u(k); \xi)$.

Taking the expectation conditioned on $u(k)$ (denoted as $\mathbb{E}_k \triangleq \mathbb{E}[\cdot|u(k)]$) we have:

$$\mathbb{E}_k[J(u(k+1))] \le J(u(k)) - \eta \left(1 - \frac{\eta\beta}{2} - \frac{\eta(1-\eta\beta)}{2\epsilon}\right) \|\nabla J(u(k))\|_2^2$$
$$+ \left(\frac{\eta^2\beta}{2} + \frac{\epsilon}{2}\right) \mathbb{E}_k[\|\nabla J(u(k)) - g(u(k); \xi)\|_2^2] \tag{14}$$

where $\mathbb{E}_k[\|\nabla J(u(k)) - g(u(k); \xi)\|_2^2]$ indicates the bias-variance balance since:

$$\mathbb{E}_k[\|\nabla J(u(k)) - g(u(k); \xi)\|_2^2] = \|\nabla J(u(k)) - \mathbb{E}_k[g(u(k); \xi)]\|_2^2 + \mathbb{E}_k[\|g(u(k); \xi) - \mathbb{E}_k[g(u(k); \xi)]\|_2^2] \tag{15}$$

The first term is the norm of bias between true gradient $\nabla J(u(k))$ and the expected gradient $\mathbb{E}_k[g(u(k); \xi)]$, while the second term represents the variance of the gradient. Substitute $g(u(k); \xi)$ by $\hat{g}_h$ we have:

$$\mathbb{E}_k[\|\nabla J(u(k)) - g(u(k); \xi)\|_2^2] = \|\mathbb{E}_k[(1 - \lambda_h)(a_h g_h - g_T)]\|_2^2$$
$$+ \text{tr}\left(\frac{a_h^2}{B}C_h + \frac{\lambda_h^2}{B'}\left(a_h^2 C_h + C_T - a_h C_{h,T}\right)\right), \tag{16}$$

which is a quadratic equation for both $a_h, \lambda_h$. Optimizing Eq. (16) over $a_h$ and $\lambda_h$, we obtain:

$$a_h^* = \frac{b}{b+d}a_{h,bias}^* + \frac{d}{b+d}a_{h,var}^*, \tag{17}$$

$$\lambda_h^* = \frac{\|\mathbb{E}_k[a_h g_h - g_T]\|_2^2}{\frac{1}{B'}\text{tr}(a_h{}^2 C_h + C_T - a_h C_{h,T}) + \|\mathbb{E}_k[a_h g_h - g_T]\|_2^2} = \frac{\text{Bias}}{\text{Var} + \text{Bias}}. \tag{18}$$

where $a_{h,bias}^*, a_{h,var}^*$ are the optimums of the bias and variance respectively, and $b, d$ are defined as follows:

$$a_{h,bias}^* = \frac{\mathbb{E}_k[g_h{}^\top g_T]}{\|\mathbb{E}_k[g_h]\|_2^2} = \underset{a}{\arg\min} \|\mathbb{E}_k[(1 - \lambda_h)(a_h g_h - g_T)]\|_2^2$$

$$a_{h,var}^* = \frac{\lambda_h^2 B}{\lambda_h^2 B + B'} \frac{\text{tr}(C_{h,T})}{\text{tr}(C_h)} = \underset{a}{\arg\min}\, \text{tr}\left(\frac{a_h^2}{B}C_h + \frac{\lambda_h^2}{B'}\left(a_h^2 C_h + C_T - a_h C_{h,T}\right)\right) \tag{19}$$

$$b = (1 - \lambda_h)^2 \|\mathbb{E}_k[g_h]\|_2^2 \ge 0, d = \left(\frac{1}{B} + \frac{\lambda_h^2}{B'}\right)\text{tr}(C_h) \ge 0.$$

From Eq. (17), we observe directly that the $a_h^*$ is the linear combination of the optimum of bias and variance, which strikes a balance between them. The weights $b, d$ are related to the bias and the variance of the local gradient $g_h$. In the early training stage where the gradient is large and so is $b$, $a_h^*$ will tend to $a_{h,bias}^*$ to reduce the bias. When the variance dominates, $d$ is larger and leads $a_h^*$ to $a_{h,var}^*$ to reduce the variance.

As for $\lambda_h^*$, since it has a similar $\frac{\text{Bias}}{\text{Bias}+\text{Variance}}$ structure as Eq. (13), we can derive the same observations: both $\lambda_h = 0$ and $\lambda_h = 1$ are sub-optimal and $\lambda_h$ can adjust the bias-variance balance for fixed $a_h$.

**Remark 4.2** *Simultaneously solving $a_h{}^*$ and $\lambda_h^*$ results in a quintic function, making the solution highly complex. However, since Eq. (15) is convex in both $a_h$ and $\lambda_h$, we can simply optimize $a_h$ and $\lambda_h$ alternatively till their convergence.*

In conclusion, the balancing of bias and variance is the crucial point in the BP-modified local loss since it reduces the bias of the local gradient at the expense of introducing additional variance. This balance exists throughout the whole analysis. If the variance is large, the estimation of the offset is inaccurate, leading the method to fail. Conversely, the balancing of bias and variance gives a useful way to sign the scaling factors.

---

**Algorithm 1** BP-modified Local Loss Algorithm

---

**Require:** Starting point $u_0$, Local loss gradient estimator $g_{h,t}$, Global loss gradient estimator $g_{T,t}$, Total training step $\mathcal{T}$, Gradient-based Optimizer Opt, Batch size $B, B'$, Sampling period $K$, Function $M$ to get $a, \lambda$
 1: $u(0) = u_0$
 2: $k = 0$
 3: **for** Training period $i \in \{1, \cdots, \lceil \frac{\mathcal{T}}{K} \rceil\}$ **do**
 4:     Sample small-batches $\mu_{B'}$ with size $B'$             $\triangleright$ Lazy update of $\Delta \tilde{g}_{h,t}, a_{h,t}, \lambda_{h,t}$
 5:     **for** Layer $t \in \{0, \cdots, T-1\}$ **do**
 6:         Compute $\tilde{g}_{h,t}, \tilde{g}_{T,t}, C_{h,t}, C_{T,t}, C_{h,T,t}$ on $\mu_{B'}$
 7:         Compute $a_{h,t}, \lambda_{h,t} = M(\tilde{g}_{h,t}, \tilde{g}_{T,t}, C_{h,t}, C_{T,t}, C_{h,T,t})$     $\triangleright$ e.g. Eq. (17), Eq. (18)
 8:         Compute $\mathbb{E}_{\mu_{B'}}[\Delta \tilde{g}_{h,t}] = \mathbb{E}_{\mu_{B'}}[\tilde{g}_{T,t} - a_{h,t}\tilde{g}_{h,t}]$           $\triangleright$ Eq. (5)
 9:     **end for**
10:     **while** $k < i * K$ and $k < \mathcal{T}$ **do**
11:         Sample small-batches $\mu_B$ with size $B$
12:         **for** Layer $t \in \{0, \cdots, T-1\}$ **do**
13:             Compute $g_{h,t}$ on $\mu_B$                 $\triangleright$ get local gradient, Eq. (4)
14:             Compute $\hat{g}_{h,t} = a_{h,t}\mathbb{E}_{\mu_B}[g_{h,t}] + \lambda_{h,t}\mathbb{E}_{\mu_{B'}}[\Delta \tilde{g}_{h,t}]$   $\triangleright$ modify local gradient, Eq. (5)
15:             Update weight $u_t(k) = \text{Opt}(u_t(k-1), \hat{g}_{h,t})$
16:         **end for**
17:         $k = k + 1$
18:     **end while**
19: **end for**
20: **return** Final weight $u(\mathcal{T})$

---

## 5   Implementation of the BP-modified Local Loss Algorithm

In this section, we present the detailed implementation of the BP-modified local loss algorithm. Instead of simply applying Eq. (5) in every step, we use the following two techniques to further mitigate the impact on both time and memory: (1) lazy update of the offset $\Delta \tilde{g}_{h,t}$ and scaling factors $a_h, \lambda_h$; (2) split mini-batch $B'$ into smaller mini-batches to compute the offset. The final algorithm is presented in Algorithm 1.

**Lazy update reducing computation overhead**   Assuming the gradient does not change rapidly, we can periodically update the offset and scaling factors and use the same values within each period to reduce the time required for computing the offset. In other words, we sample $\mu_{B'}$ and compute the offset $\Delta g_h$ and scaling factors $a_h, \lambda_h$ at the beginning of every $K$ steps, and apply the same offset in Eq. (5) to modify the local gradient in the next $K$ steps. The additional relative computation overhead will be about $\frac{(1+r)B'}{KB}$ where $r$ is the ratio of the computation time of computing the true gradient to the computation time of acquiring the local gradient.

**Batch split reducing memory overhead**   Further, the total memory overhead of the BP-modified local loss method will be on the scale of $O(Bh, B'h, B'T)^2$. We need $B' \lesssim \frac{h}{T}B$ to sustain comparable memory usage as the original algorithm, i.e. $O(Bh)$. However, small $B'$ will increase the additional variance introduced by the offset (Eq. (7)). Combined with the impact of lazy update strategy (see following discussion and Appendix C for details), the variance of the offset will be

---

[2]Firstly, the memory usage will increase linearly with batch size $B$ and it is reported that the memory usage will grow linearly with respect to horizon $h$ (Ren & Li, 2024), and these two factors are independent. Secondly, acquiring different gradients is asynchronous so only need to consider the maximum memory for different gradients.

unacceptable. Therefore, we further split the mini-batch $\mu_{B'}$ into smaller mini-batches and use the gradient accumulation technique to compute the offset, and the memory overhead will reduce to $O(Bh, b'T, b'h)$ where $b'$ is the size of the smaller mini-batch.

However, some problems will appear after applying these two strategies. For example, the noise in the offset will be accumulated in the period thus increasing the equivalent variance. As we use the same offset in one period, the same noise will accumulate, leading to a larger variance in the long run. For further discussion of the impact of delayed gradient adjustment, please refer to Appendix C. Secondly, further splitting $\mu_{B'}$ into smaller mini-batches requires more computation time. However, the lazy update relieves this issue. As long as $B'$ is smaller than the scale of $KB$, the impact of computation time will be small, thus $B'$ no longer needs to be smaller than $B$.

From the above discussion, we know that larger batch size $B'$ and quicker update (i.e. smaller period $K$) will reduce the variance of the offset and the delayed bias thus improving performance, at the expense of more computation. We have shown the numerical results of different choices of $K$ and $B'$ in Appendix D.2 and its result is aligned with the analysis.

There is an inner loop (i.e. the update period) in the implemented BP-modified local loss method, and $B'$ will become larger by introducing the lazy update. This procedure is similar to control variate methods like SVRG (Johnson & Zhang, 2013), where an additional term is introduced (the variance reduction term in SVRG and the offset $\Delta \tilde{g}_h$ in our method), which is computation-consuming and updated periodically. However, the basic motivation of the control variate method is to reduce the variance, while our method aims to reduce the bias. Furthermore, the variance reduction methods are based on acquiring the true gradient, i.e., BP, while our method is based on the local loss gradient.

## 6 NUMERICAL RESULTS

To evaluate the effectiveness of the proposed BP-modified local loss method, we conducted several experiments on CIFAR100 (Krizhevsky, 2009) and ImageNet-Tiny (mnmoustafa & Ali, 2017) dataset: (1) Fine-tuning a pre-trained ResNet-50 model (He et al., 2016); (2) LoKr fine-tuning (Hyeon-Woo et al., 2022) of a pre-trained ResNet-50; (3) LoRA fine-tuning (Hu et al., 2022) pre-trained ViT-b16 model. We test the proposed method on these tasks with Forward-Forward algorithm (FF) (Hinton, 2022), LoCo algorithm (Xiong et al., 2020), and MPC algorithm (Ren & Li, 2024) with horizon $h = 5$ and the traditional BP algorithm. All the experiments are done by ourselves since no related results are provided in the original papers.

All the experiments use batch size $B = 64$, learning rate 0.001 and epoch 30, and SGD optimizer with momentum 0.9. For the BP-modified local loss method, the batch size to compute the offset is $B' = 320$ and split into mini-batches with size 8, and the period of the offset update is one epoch, i.e. $K = \lceil \frac{N}{B} \rceil$, where $N$ is the size of the training dataset. The input image is resized to (224,224) and then normalized without any other enhancement before training. For the LoRA and LoKr, the rank and alpha are set to be $r = 1, \alpha = 4$. All the experiments were conducted using a single NVIDIA GeForce RTX 3090 GPU using PyTorch (Paszke et al., 2019).

In summary, the numerical experiments show that

1. The BP-modified local loss method outperforms the original local loss in terms of performance.

2. The performance improvement is more significant in small horizons, i.e. Forward-Forward algorithm and LoCo algorithm.

3. Ablation studies on the impact of the offset show the importance of the offset in improving the local loss.

We further provide more experiment results in Appendix D including: (1) memory/time usage of the BP-modified local loss method (Appendix D.1); (2) sensitivity analysis of the update period $K$ and the offset batch size $B'$ (Appendix D.2); (3) number of epochs the BP-modified local loss method needs to achieve the original test accuracy (Appendix D.3).

Table 1: Test accuracy of BP-modified local loss on different methods and tasks on CIFAR100 Dataset. The numbers in parentheses indicate the improvement over the original local loss method.

| Method | | Task | | |
| --- | --- | --- | --- | --- |
| | | ResNet-50 Full Tune | ResNet-50 LoKr | ViT-b16 LoRA |
| BP | | 82.90 | 76.47 | 91.49 |
| FF | original | 40.10 | 54.43 | 63.65 |
| | BP-modified | 68.30 (+28.20) | 65.34 (+10.91) | 86.30 (+22.65) |
| LoCo | original | 45.29 | 55.84 | 74.31 |
| | BP-modified | 74.61 (+29.32) | 66.15 (+10.31) | 90.54 (+16.23) |
| MPC (h=5) | original | 66.02 | 65.65 | 84.03 |
| | BP-modified | 77.24 (+11.22) | 67.77 (+2.12) | 90.29 (+6.26) |

## 6.1 PERFORMANCE IMPROVEMENT OF THE BP-MODIFIED LOCAL LOSS METHOD

In this section, we highlight the performance of the BP-modified local loss method in improving local loss. We compare the results of the BP-modified method against the original methods on the same model and tasks with identical training settings.

Table 1 shows the test accuracy of the BP-modified local loss method on different methods and tasks. The results indicate that the BP-modified local loss method significantly outperforms the original local loss method across all tasks and methods. It achieves an average improvement of 20.53% on the FF method, 15.29% on the LoCo method, and 6.53% on the MPC method with horizon $h = 5$. The test accuracy of the BP-modified local loss method is comparable to the full BP method.

Additionally, we observed that the performance improvement is more significant in the FF method than in the LoCo and MPC methods. Since the FF method is the MPC method with horizon $h = 1$, this result suggests that the BP-modified local loss method is more effective when the original performance is worse. As the FF method uses the least global information, the bias of the local gradient is more significant, leading to a larger offset as shown in Eq. (18) thus a more significant performance improvement.

Further, we observed that the test accuracy of the BP-modified local loss method is higher in larger horizons, i.e. LoCo and MPC $h = 5$ cases. These original local loss methods already perform better, indicating a smaller bias and maybe more alignment with the true gradient. This results in a smaller variance of the offset (Eq. (7)). However, the performance improvement of the BP-modified local loss method in the LoKr task with horizon $h = 5$ is much smaller than the other tasks. This is largely due to the estimated variance dominating the bias-variance trade-off, leading to a smaller offset.

We also conducted experiments on the ImageNet-Tiny dataset with the same models and settings used in the CIFAR100 dataset, except we set the total epochs to 10 and period $K = 500$. The results are shown in Table 2. Consistent results are found that the BP-modified local loss method can improve the original method when the original performance is worse. In the LoKr tuning case where the original methods have achieved good performance, then the proposed method cannot improve their performance.

We also provide the memory and time usage of the above experiments in Appendix D. The results show that the memory and time usage of the BP-modified local loss method are comparable to the original local loss method. Additionally, the memory overhead is significantly lower than the full BP method.

## 6.2 ABLATION STUDY ON THE IMPACT OF THE OFFSET

In the ablation study, we investigate the impact of the offset on the performance of the BP-modified local loss method. We set $\lambda_h = 0$ to remove the offset from the BP-modified local loss method, leaving the scaling factor $a_h = a_{h,bias}^*$. We conducted the study on the FF method and show the results in Table 3. Although the scaling factor $a_h$ can improve the performance of the original local loss method without the offset, adding the offset can further improve the performance. This result indicates the importance of both the offset and the scaling factor in improving the local loss method.

Table 2: Test accuracy of BP-modified local loss on different methods and tasks on ImageNet-Tiny Dataset. The numbers in parentheses indicate the improvement over the original local loss method.

| Method | | Task | | |
|---|---|---|---|---|
| | | ResNet-50 Full Tune | ResNet-50 LoKr | ViT-b16 LoRA |
| BP | | 78.56 | 74.22 | 87.31 |
| FF | original | 24.58 | 61.23 | 55.03 |
| | BP-modified | 68.14 (+43.56) | 53.58 (-7.65) | 75.92 (+20.89) |
| LoCo | original | 31.02 | 61.97 | 58.55 |
| | BP-modified | 68.49 (+37.47) | 66.50 (+4.93) | 74.13 (+15.58) |
| MPC (h=5) | original | 55.11 | 66.27 | 72.18 |
| | BP-modified | 66.03 (+10.92) | 65.30 (-0.97) | 76.48 (+4.30) |

Table 3: Test accuracy of the BP-modified local loss method with and without the offset on the FF method. The numbers in parentheses indicate the improvement over the original local loss method.

| Method | | Task | | |
|---|---|---|---|---|
| | | ResNet-50 Full Tune | ResNet-50 LoKr | ViT-b16 LoRA |
| BP | | 82.90 | 76.47 | 91.49 |
| FF | original | 40.10 | 54.43 | 63.65 |
| | BP-modified ($\lambda = 0$) | 46.21 (+6.11) | 59.71 (+5.28) | 74.36 (+10.71) |
| | BP-modified | 68.30 (+28.20) | 63.71 (+9.28) | 86.30 (+22.65) |

Further sensitivity experiments on the update period $K$ and the offset batch size $B'$ are provided in Appendix D.2.

## 7 DISCUSSION

In this paper, we introduced a novel BP-modified local loss method. This method incorporates an additional offset to adjust the local gradient. Through the use of the stochastic modified equation, we demonstrated that the offset helps reduce the bias between the local gradient and the true gradient. However, it introduces additional variance. By employing the Ornstein-Uhlenbeck (OU) process and one-step loss analysis, we analyzed the bias-variance trade-off and derived optimal scaling factors, $\lambda_h$ and $a_h$. The impact of delayed versions of the offset was also explored. Numerical experiments across various models, tasks, and optimizers illustrate the potential of the proposed method. However, certain limitations should be acknowledged.

The OU process model used in Section 4.1 is a simplified approximation of the true stochastic gradient descent (SGD) dynamics in neural network training. Its primary role is to provide insight into the global bias-variance trade-off in the training dynamics. While several simplifications were made to obtain an explicit solution, the OU model is sufficient to capture the key phenomena of bias-variance balance and the sub-optimality of the unbiased modification. These insights were later validated in our one-step analysis and numerical experiments.

Some important aspects, however, remain under-explored. For instance, the bias introduced by the delayed offset, as discussed in Section C, and certain hyperparameters, such as the offset batch size $B'$ and the sampling period $K$, are not fully analyzed. Although these factors are critical, their complexity exceeds the scope of this work and is left for future investigation.

The computation of the offset requires access to the true gradient. This process involves further splitting the batch size $B'$ into smaller sub-batches, as discussed in Section 5. This adds complexity to the training process. Integrating this approach with methods like forward gradient techniques (e.g., MeZO (Malladi et al., 2023)) and control variate gradient estimation (e.g., (Arisaka & Li, 2024)) could be a promising direction. We reserve this for future exploration.

Due to limited computational resources, we did not experiment with larger models or datasets, and all experiments were conducted on a single GPU without distributed implementation. Nevertheless, our experiments across various models and tasks consistently demonstrated the potential of the

BP-modified local loss method. Further optimization of the method and implementation, including distributed settings and tests on larger datasets, will be explored in future work.

ACKNOWLEDGMENTS

This research is supported by the National Research Foundation, Singapore, under the NRF fellowship (project No. NRF-NRFF13-2021-0005).

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

## A  EXPLANATION FOR LOSS $L$ (REN & LI, 2024)

In Ren & Li (2024), the author states that the loss function $L$ includes the head block and other possible weights, and target $y$ is omitted as stated in the main paper. Moreover, it can vary through the block, i.e. the loss function $L$ in $L(x_t)$ and $L(x_{t+1})$ in Eq. (2) can be different. However, in this paper, we still assume the loss needs to have the same structure.

For example, assuming an image classification task, it is allowed that the two loss functions $L_t$ in $L_t(x_t)$ and $L_{t+1}(x_{t+1})$ with the same global pooling-linear-softmax-cross-entropy loss structure to have independent weights in its linear layer and the weight even size of the linear layer can be different (e.g., in CNN networks, the number of channels will increase through the block). We still think of them as the 'same' loss and thus do not distinguish them in the main paper. However, it is not allowed that $L_t$ use cross-entropy loss while $L_{t+1}$ use contrastive loss.

The study of using different losses in different blocks is left for future work.

## B  DETAILED DERIVATIONS

In this section, we will give detailed derivations of Section 4.

### B.1  DERIVATIONS OF THE STOCHASTIC MODIFIED EQUATIONS (6) AND (7)

In this section, we will give the derivation of the stochastic modified equation Eq. (6) and Eq. (7) using the analysis and result of Li et al. (2017).

For the SGD update of a general gradient estimator $g(u(k); \xi(k))$, where the random variable $\xi(k)$ controls its randomness:

$$u(k + 1) = u(k) - \eta g(u(k); \xi(k)) = u(k) - \eta \mathbb{E}_{\xi(k)}[g(u(k); \xi(k))] + \sqrt{\eta}V(u(k); \xi(k)), \quad (20)$$

where $\eta$ is the learning rate and $V(u(k); \xi(k)) = \sqrt{\eta}\mathbb{E}_{\xi(k)}[g(u(k); \xi(k))] - \sqrt{\eta}g(u(k); \xi(k))$. By simple computation, we can know:

$$\mathbb{E}_{\xi(k)}[V(u(k); \xi(k))] = 0, Cov(V(u(k); \xi(k))) = \eta Cov(g(u(k); \xi(k))). \quad (21)$$

Thus Eq. (20) can be seen as the Euler-Maruyama discretization of the SDE:

$$dU(\tau) = -b(U, \tau)d\tau + \sqrt{\eta \Sigma(U, \tau)}dW_\tau \quad (22)$$

for a time step $\delta_\tau = \eta$ and $u(k) \approx U(\eta k)$, where $b(U, \tau) = \mathbb{E}_{\xi(\tau)}[g(U(\tau); \xi(\tau)], \Sigma(U, \tau) = Cov(g(U(\tau); \xi(\tau))$.

Denoting $G$ is the set of functions that have at most polynomial growth, i.e.

$$g \in G \Leftrightarrow \exists \kappa_1, \kappa_2, L > 0, s.t. \|g(x)\|_2^2 \le L(\kappa_1 + \|x\|_2^{2\kappa_2}), \quad (23)$$

and $G^n$ is the set of $n$-times continuously differentiable functions with its partial derivatives belonging in $G$, and $G_w^n$ is the weak differentiable analog (detailed definition please refer to (Li et al., 2017)). From Theorem 9 and Corollary 10 in (Li et al., 2017), if the gradient estimator $g(u(k); \xi(k))$ satisfies the following assumptions:

1. Uniform linear growth for mean and variance in $U, \tau$, i.e.

$$\|b(U, \tau)\|_2^2 + \|\Sigma(U, \tau)\|_F^2 \le \beta_1(1 + \|U\|_2^2), \forall U \in \mathbb{R}^m, \tau \in [0, 1] \quad (24)$$

   for some $\beta_1 > 0$,

2. Uniform Lipschitz condition for mean and variance in $U, \tau$, i.e.

$$\|b(U_1, \tau) - b(U_2, \tau)\|_2^2 + \|\Sigma(U_1, \tau) - \Sigma(U_2, \tau)\|_F^2 \le \beta_2\|U_1 - U_2\|_2^2, \forall U \in \mathbb{R}^m, \tau \in [0, 1], \quad (25)$$

3. $b(U, \tau)$ is continuously differentiable and $b(U, \tau) \in G_w^3$,

4. $\nabla g(U(\tau), \xi(\tau))$ is Lipschitz continuous almost surely in $U$ with Lipschitz constant $\beta_{\xi(\tau)}$ and $\mathbb{E}_{\xi(\tau)}[\beta_{\xi(\tau)}^m] < \infty$ for all $\tau \in [0, 1]$ and $m \ge 1$,

then $U(\tau)$ is a 1-order weak approximation of $u(k)$, i.e. for each $g \in G^2$, exists $C > 0$ independent of $\eta$, such that:

$$\max_{k=0,\cdots,\mathcal{T}} \|\mathbb{E}_{\xi(\tau)}[u(k) - U(\eta k)]\|_2 \le C\eta \tag{26}$$

Then we just substitute $g(u; \xi)$ by local gradient $g_h$ to get the Eq. (6). As for Eq. (7), the mean and covariance modified gradient $\hat{g}_h$ are:

$$\mathbb{E}_\mu[\hat{g}_h] = \mathbb{E}_\mu[(1 - \lambda_h)a_h g_h + \lambda_h g_T] \tag{27}$$

$$Cov(\hat{g}_h) = \frac{a_h^2}{B} Cov(g_h) + \frac{\lambda_h^2}{B'} Cov(\Delta \tilde{g}_h). \tag{28}$$

Then we can get the 1-order stochastic modified equations of the proposed method Eq. (7).

## B.2 DERIVATION OF OU PROCESS

In this section, we will derive the Eq. (12) and Eq. (13) in Section 4.1.

By Eq. (10):

$$dU_h(\tau) = -\nabla J_h(U_h(\tau))d\tau + \sqrt{V}dW_t = -A(U_h(\tau) - u_h^*)d\tau + \sqrt{V}dW_t, \tag{29}$$

and the property of the OU process, the distribution of $U_h(\mathcal{T})$ satisfies the Gaussian distribution:

$$U_h(\mathcal{T}) \sim \mathcal{N}\left(e^{-A\mathcal{T}}u_0 + u_h^*(\mathcal{T}), \Sigma_{\mathcal{T}}\right) \tag{30}$$

where $u_h^*(\mathcal{T}) = (I - e^{-A\mathcal{T}})u_h^*$, and $\Sigma_{\mathcal{T}} = \int_0^{\mathcal{T}} e^{-At}Ve^{-A^\top t}\,d\tau$. The expectation loss at time $\mathcal{T}$ is:

$$\mathbb{E}[J(U_h(\mathcal{T}))] = \frac{1}{2}\mathrm{tr}(\Sigma_T A) + J(\mathbb{E}[U_h(\mathcal{T})]). \tag{31}$$

which consist of a bias-related term $J(\mathbb{E}[U_h(\mathcal{T})])$ and a variance-related term $\frac{1}{2}\mathrm{tr}(\Sigma_{\mathcal{T}}A)$.

Similarly by Eq. (11):

$$d\hat{U}_h(\tau) = -((1 - \lambda_h)A(\hat{U}_h(\tau) - u_h^*) + \lambda_h A\hat{U}_h(\tau))d\tau + \sqrt{V + \lambda_h^2 \tilde{V}}dW_t, \tag{32}$$

the distribution of $\hat{u}_h(\mathcal{T})$ is:

$$\hat{u}_h(\mathcal{T}) \sim \mathcal{N}\left(e^{-A\mathcal{T}}u_0 + (1 - \lambda_h)u_h^*(\mathcal{T}), \Sigma_{\mathcal{T}} + \lambda_h^2 \Sigma_{\mathcal{T}}'\right) \tag{33}$$

where $\tilde{\Sigma}_{\mathcal{T}} = \lambda_h^2 \int_0^{\mathcal{T}} e^{-At}\tilde{V}e^{-A^\top t}\,d\tau$. The expected loss is:

$$\mathbb{E}[J(\hat{U}_h(\mathcal{T}))] = \mathbb{E}[J(U_h(\mathcal{T}))] + \frac{\lambda_h^2}{2}\left(\mathrm{tr}(\Sigma_{\mathcal{T}}'A) + \|u_h^*(\mathcal{T})\|_A^2\right) - \lambda_h \mathbb{E}[U_h(\mathcal{T})]^\top Au_h^*(\mathcal{T}). \tag{34}$$

To minimize the expectation loss, we need to balance the bias and variance by $\lambda_h$, which results in:

$$\lambda_h^*(\mathcal{T}) = \frac{(e^{-A\mathcal{T}}u_0)^\top Au_h^*(\mathcal{T}) + \|u_h^*(\mathcal{T})\|_A^2}{\mathrm{tr}(\tilde{\Sigma}_{\mathcal{T}}A) + \|u_h^*(\mathcal{T})\|_A^2}, \tag{35}$$

$$\min_{\lambda_h} \mathbb{E}[J(\hat{U}_h(\mathcal{T}))] = \mathbb{E}[J(U_h(\mathcal{T}))] - \frac{\left(\mathbb{E}[U_h(\mathcal{T})]^\top Au_h^*(\mathcal{T})\right)^2}{2\left(\mathrm{tr}(\tilde{\Sigma}_{\mathcal{T}}A) + \|u_h^*(\mathcal{T})\|_A^2\right)}, \tag{36}$$

which is Eq. (12) and Eq. (13)

## B.3 DERIVATIONS IN ONE-STEP LOSS ANALYSIS

In this section, we will derive Eq. (14) and (16) in Section 4.2

By the $\beta-$Lipstichz continuity of the gradient $\nabla J$, we have:

$$J(u_1) - J(u_2) = \int_0^1 \nabla J(tu_1 + (1 - t)u_2)^\top (u_1 - u_2)\,dt$$

$$= \nabla J(u_2)^\top (u_1 - u_2) + \int_0^1 (\nabla J(tu_1 + (1 - t)u_2) - \nabla J(u_2))^\top (u_1 - u_2)\,dt$$

$$\le \nabla J(u_2)^\top (u_1 - u_2) + \frac{\beta}{2}\|u_1 - u_2\|_2^2 \tag{37}$$

Substitute $u_1, u_2$ by $u(k+1), u(k)$, we have:

$$J(u(k+1)) - J(u(k))$$

$$\leq \nabla J(u(k))^\top (u(k+1) - u(k)) + \frac{\beta}{2}\|u(k+1) - u(k)\|_2^2$$

$$= -\eta \nabla J(u(k))^\top g(u(k);\xi) + \frac{\eta^2 \beta}{2}\|g(u(k);\xi)\|_2^2 \tag{38}$$

$$= -(1-\eta\beta)\eta \nabla J(u(k))^\top g(u(k);\xi) + \frac{\eta^2\beta}{2}(\|g(u(k);\xi) - \nabla J(u(k))\|_2^2 - \|\nabla J(u(k))\|_2^2)$$

$$\leq -\eta\left(1 - \frac{\eta\beta}{2} - \frac{\eta(1-\eta\beta)}{2\epsilon}\right)\|\nabla J(u(k))\|_2^2 + \left(\frac{\eta^2\beta}{2} + \frac{\epsilon}{2}\right)\|\nabla J(u(k)) - g(u(k);\xi)\|_2^2$$

where the second inequality is the Cauchy-Schwartz inequality for arbitrary $\epsilon > 0$. Taking expectation condition on $u(k)$ on both sides we can get Eq. (14):

$$\mathbb{E}_k[J(u(k+1))] \leq J(u(k)) - \eta\left(1 - \frac{\eta\beta}{2} - \frac{\eta(1-\eta\beta)}{2\epsilon}\right)\|\nabla J(u(k))\|_2^2$$

$$+ \left(\frac{\eta^2\beta}{2} + \frac{\epsilon}{2}\right)\mathbb{E}_k[\|\nabla J(u(k)) - g(u(k);\xi)\|_2^2] \tag{39}$$

For the derivation of $a_h^*, \lambda_h^*$, consider the last term in the previous equation (i.e. $\mathbb{E}_k[\|\nabla J(u(k)) - g(u(k);\xi)\|_2^2]$) and substitute $g(u(k);\xi)$ by $\hat{g}_h$, we have:

$$\mathbb{E}_k[\|\nabla J(u(k)) - g(u(k);\xi)\|_2^2]$$

$$= \|\nabla J(u(k)) - \mathbb{E}_k[g(u(k);\xi)]\|_2^2 + \mathbb{E}_k[\|g(u(k);\xi) - \mathbb{E}_k[g(u(k);\xi)]\|_2^2]$$

$$= \|\mathbb{E}_k[(1-\lambda_h)(a_h g_h - g_T)]\|_2^2 + \text{tr}\left(\frac{a_h^2}{B}C_h + \frac{\lambda_h^2}{B'}\tilde{C}_h\right) \tag{40}$$

$$= \|\mathbb{E}_k[(1-\lambda_h)(a_h g_h - g_T)]\|_2^2 + \text{tr}\left(\frac{a_h^2}{B}C_h + \frac{\lambda_h^2}{B'}\left(a_h^2 C_h + C_T - a_h C_{h,T}\right)\right),$$

which is Eq. (16)

## C   IMPACT OF DELAYED GRADIENT ADJUSTMENT

Since the computation of gradient adjustment in the BP-modified local loss method every step can be computationally expensive, in practice, we update the sample offset $\Delta g_h$ and $a_h, \lambda_h$ only every $K$ steps, which reduces the computational cost. This delayed gradient adjustment introduces two effects on the dynamics: noise accumulation and biased adjustment.

**Accumulation of Noise**   Applying the same offset noise over multiple steps in a period introduces a higher variance. To illustrate this, consider the OU process in Section 4.1, using the Euler-Maruyama method with timestep $dt$ and $A = I, \xi_\tau \sim N(0, Vdt), \tilde{\xi}_\tau \sim N(0, \lambda_h^2 \tilde{V}dt)$, then the update are:

$$u_h(\tau + dt) = u_h(\tau) - dt(u_h(\tau) - u_h^*) + \xi_\tau, \tag{41}$$

$$\tilde{u}_h(\tau + dt) = \tilde{u}_h(\tau) - dt(\tilde{u}_h(\tau) - u_h^*) + \xi_\tau - dtu_h^* + \tilde{\xi}_\tau = (I - dt)\tilde{u}_h(\tau) + \xi_\tau + \tilde{\xi}_\tau. \tag{42}$$

After $K$ steps, the solutions are:

$$u_h(Kdt) = (1-dt)^K u_0 + (1 - (1-dt)^K)u_h^* + \sum_{\tau=0}^{K-1}(1-dt)^{K-\tau-1}\xi_\tau, \tag{43}$$

$$\tilde{u}_h(Kdt) = (1-dt)^K u_0 + \sum_{\tau=0}^{K-1}(1-dt)^{K-\tau-1}\xi_\tau + \sum_{\tau=0}^{K-1}(1-dt)^{K-\tau-1}\tilde{\xi}_\tau. \tag{44}$$

If we apply the first modify term $-u_h^* dt + \tilde{\xi}_0$ over the entire $K$ steps, the solution for $\tilde{u}_h(Kdt)$ becomes:

$$\tilde{u}_h(Kdt) = (1-dt)^K u_0 + \sum_{\tau=0}^{K-1}\xi_\tau(1-dt)^{K-\tau-1} + \frac{1 - (1-dt)^K}{dt}\tilde{\xi}_0. \tag{45}$$

The variance of the noise terms can be computed as:

$$\sum_{\tau=0}^{K-1}(1-dt)^{K-\tau-1}\xi'_\tau \sim \mathcal{N}\left(0, \frac{1-(1-dt)^{2K}}{2dt+dt^2}\lambda_h^2\tilde{V}\right), \frac{1-(1-dt)^K}{dt}\tilde{\xi}_0 \sim \mathcal{N}\left(0, \left(\frac{1-(1-dt)^K}{dt}\right)^2\lambda_h^2\tilde{V}\right).$$
$$(46)$$

When $dt \ll 1$, we have $\frac{1-(1-dt)^{2K}}{2dt+dt^2} \sim K$ and $(\frac{1-(1-dt)^K}{dt})^2 \sim K^2$, indicating that the variance increases by a factor of approximately $K$. This noise accumulation will also happen in the stochastic modified equations (Eq. (7)), and thus in practice, we multiply the variance introduced by the offset by a factor of $K$ (i.e. $\frac{\eta\lambda_h^2 K}{B'}\tilde{C}_h$).

**Biased Adjustment** The delayed gradient adjustment introduces additional bias $\mathbb{E}[\lambda_h a_h(g_h(\tau) - g_h(\tau')) - \lambda_h(g_T(\tau) - g_T(\tau'))]$ where $\tau'$ is the last offset-sampling step. As mentioned in Remark 4.1, when $\lambda_h = 1$, the modified gradient is an unbiased estimator of the true gradient. However, in the delayed adjustment case, this is no longer true. This additional bias introduced by this delay will also affect the bias-variance trade-off and thus the optimal values of $a_h$ and $\lambda_h$, which we leave for future study.

## D    FURTHER EXPERIMENT RESULTS

This section provides additional experimental results to further demonstrate the effectiveness of the BP-modified local loss method. We report the memory usage and runtime of the BP-modified local loss method compared to the original local loss and full BP methods of the experiments in Section 6.1. We also provide additional results on the impact of period $K$ and batch size $B'$ on the performance and efficiency of the BP-modified local loss method.

### D.1    COMPARISON OF MEMORY USAGE AND RUNTIME

We compared the memory usage and runtime of the BP-modified local loss method against the original local loss and full BP methods. Memory usage was evaluated by profiling several training steps, including the original training step and the computation of offsets, and recording peak memory usage. Detailed procedures for memory profiling are provided in Appendix E.

Table 4 indicates that the memory usage of the BP-modified local loss method is comparable to the original local loss method, which is significantly lower than the full BP method. The additional memory required by the BP-modified method is minimal, as it primarily involves storing offsets (equal to the size of trainable parameters) and scalar scaling factors. As described in Section 5, most memory usage during training stems from storing intermediate states for backpropagation, making the memory overhead introduced by the BP-modified method negligible. This is particularly evident in low-rank methods like LoRA and LoKr, where the trainable parameter size is significantly smaller than the model size, rendering the memory overhead negligible.

On the other hand, Table 5 compares the runtime per epoch of the original and BP-modified methods. The BP-modified local loss method incurs a small runtime increase due to the computation of offsets and scaling factors. This overhead is mitigated by the lazy update strategy. In our experimental setup with $B' = 320$ and period $K = 1\ epoch$, the additional runtime is under 6% of the original local loss method. This increase is higher than the expected batch size ratio ($\frac{B'}{N} = 0.64\%$) due to the mini-batch split technique.

Combining memory and runtime efficiency with performance results from the previous section, the BP-modified local loss method demonstrates a strong balance of accuracy and resource usage.

### D.2    SENSITIVITY TO PERIOD $K$ AND MODIFY BATCH SIZE $B'$

We further examine the sensitivity of the BP-modified local loss method to the update period $K$ and modify batch size $B'$ in the BP-modified local loss method. In the FF algorithm experiments, we vary $K = \{0.1\ epoch, 0.3\ epoch, 1\ epoch\}$ and $B' = \{32, 320, 3200\}$, while keeping other hyperparameters consistent with the settings in Section 6.

Table 4: Memory usage (MB) of BP-modified local loss on different methods and tasks. The numbers in parentheses indicate the additional memory usage and its percentage of the BP-modified local loss method compared to the original local loss method

| Method | | Task | | |
|---|---|---|---|---|
| | | ResNet-50 Full Tune | ResNet-50 LoKr | ViT-b16 LoRA |
| BP | | 5617 | 5529 | 5586 |
| FF | original | 1555 | 1366 | 1519 |
| | BP-modified | 1642 (+88 (5.63%)) | 1370 (+4.1 (0.30%)) | 1520 (+0.15 (0.01%)) |
| LoCo | original | 2239 | 2251 | 1926 |
| | BP-modified | 2329 (+90 (4.02%)) | 2252 (+1.1 (0.05%)) | 1926 (+0.15 (0.01%)) |
| MPC (h=5) | original | 3965 | 3965 | 3146 |
| | BP-modified | 4055 (+91 (2.29%)) | 3968 (+3.1 (0.08%)) | 3146 (+0.15 (0.005%)) |

Table 5: Runtime per epoch (s) of BP-modified local loss on different methods and tasks. The numbers in parentheses indicate the additional runtime and its percentage of the BP-modified local loss method compared to the original local loss method

| Method | | Task | | |
|---|---|---|---|---|
| | | ResNet-50 Full Tune | ResNet-50 LoKr | ViT-b16 LoRA |
| BP | | 127.07 | 193.51 | 322.84 |
| FF | original | 123.03 | 171.11 | 291.99 |
| | BP-modified | 126.79 (+3.76 (3.06%)) | 177.1 (+5.99 (3.50%)) | 296.37 (+4.38 (1.50%)) |
| LoCo | original | 201.53 | 298.00 | 527.05 |
| | BP-modified | 205.67 (+4.14 (2.05%)) | 314.58 (+16.58 (5.56%)) | 515.08 (-11.97 (-2.27%)) |
| MPC (h=5) | original | 363.71 | 566.94 | 967.10 |
| | BP-modified | 367.48 (+3.77 (1.04%)) | 578.08 (+11.14 (1.96%)) | 979.28 (+12.18 (1.26%)) |

Theoretically, a larger period $K$—corresponding to a less frequent update of $\Delta g_{h,t}$ and scaling factors $a_{h,t}, \lambda_{h,t}$—introduces greater variance and amplifies the delayed bias, potentially impacting performance, as discussed in Appendix C. Conversely, a smaller $B'$ increases variance, as suggested by Eq. (7). In contrast, using a smaller $K$ or a larger $B'$ improves performance but comes at the cost of higher computational requirements.

Table 6 and 7, summarize the results for different $B'$ and $K$ values. These results align with our theoretical analysis: larger $B'$ and smaller $K$ lead to better performance but increases computation. A detailed comparison shows that if the mini-batch size $B'$ is too small, the variance of the modified gradient will be large and the performance might be worse, while it seems that it has more tolerance for larger $K$. If the period $K$ is too small or $B'$ is too large, the computation overhead will be large with diminishing performance improvement[3]. Our experience suggests that $B'$ should be large enough to reduce the variance of the modified gradient while $K$ can be relatively large such that the computation overhead is small. $B'$ being $1\%$ of $BK$ are usually good choices.

Table 6: Sensitivity of modify batch size $B'$. Test accuracy (in percentage) and training time (second per epoch) are reported.

| | Method | ResNet-50 Full Tune | | ResNet-50 LoKr | | ViT-b16 LoRA | |
|---|---|---|---|---|---|---|---|
| | | Test Acc | Time(s) | Test Acc | Time(s) | Test Acc | Time(s) |
| | original | 40.10 | 123.03 | 54.43 | 171.11 | 63.65 | 291.99 |
| | BP-modified (B'=32) | 38.97 | 125.32 | 57.84 | 173.19 | 74.88 | 317.63 |
| FF | BP-modified (B'=320) | 68.30 | 126.79 | 65.34 | 177.10 | 86.30 | 296.37 |
| | BP-modified (B'=3200) | 74.62 | 152.88 | 71.91 | 219.16 | 90.47 | 333.50 |

---

[3]Noted that there is no change in memory overhead because of batch split technique. Larger $B'$ means more computation but no more memory usage.

Table 7: Sensitivity of period $K$. Test accuracy (in percentage) and training time (second per epoch) are reported.

| | Method | ResNet-50 Full Tune | | ResNet-50 LoKr | | ViT-b16 LoRA | |
|---|---|---|---|---|---|---|---|
| | | Test Acc | Time(s) | Test Acc | Time(s) | Test Acc | Time(s) |
| | original | 40.10 | 123.03 | 54.43 | 171.11 | 63.65 | 291.99 |
| | BP-modified (K=0.1Epoch) | 73.98 | 152.30 | 72.99 | 218.52 | 89.97 | 337.07 |
| FF | BP-modified (K=0.3Epoch) | 68.18 | 133.28 | 65.34 | 186.26 | 87.83 | 324.80 |
| | BP-modified (K=1Epoch) | 68.14 | 126.79 | 65.34 | 177.10 | 86.30 | 296.37 |

Table 8: Average first epoch to reach the same test accuracy as the original local loss method in the CIFAR100 dataset. The numbers in parentheses indicate the final test accuracy of the original methods.

| | Task | | |
|---|---|---|---|
| Method | ResNet-50 Full Tune | ResNet-50 LoKr | ViT-b16 LoRA |
| FF | 1.0 (40.10) | 3.0 (54.43) | 4.0 (63.65) |
| LoCo | 1.0 (45.29) | 3.3 (55.84) | 2.0 (74.31) |
| MPC (h=5) | 2.0 (66.02) | 13.0 (65.65) | 4.0 (84.03) |

### D.3 EPOCHS TO GET THE ORIGINAL PERFORMANCE

Here we provide the number of epochs needed for the BP-modified local loss method to achieve the same test accuracy as the original local loss method in CIFAR100 dataset. We report the results in Table 8. We find that the BP-modified local loss method requires much less epochs to reach the same performance as the original local loss method, demonstrates the efficiency of the proposed method. It is also consistent with the significant improvement in test accuracy.

## E  RECORDING THE MEMORY USAGE

We did not directly check the GPU memory usage by PyTorch, instead, we used the *torch.profiler* module to capture the GPU memory allocation by torch following this blog[4]. This method can show detailed memory usage through training steps. We construct the same model as training and call the training procedure for a few steps including some offset computation steps. Then we use the peak memory allocation as the memory usage of that method to train the model. Each experiment was done three times and reported their average.

---

[4]https://pytorch.org/blog/understanding-gpu-memory-1/

