# OpenReview forum: "BP-Modified Local Loss for Efficient Training of Deep Neural Networks"
_ICLR.cc/2025/Conference — ICLR 2025 Poster_

### Official Review · Reviewer_UZvZ · 2024-10-16

**Soundness:** 3
**Presentation:** 4
**Contribution:** 3
**Rating:** 8
**Confidence:** 4

**Summary:**

This paper proposes a novel BP-modified local loss to improve the performance of local loss training, which decreases the bias between the BP gradient and local loss gradient but introduces additional variance. Then authors provide a theoretical analysis using the stochastic modified equation, illustrating the bias-variance trade-off and deriving optimal scaling factors. Experiment results show that the proposed method can effectively improve the performance of local loss algorithm while increase little GPU memory.

**Strengths:**

- The theoretical derivation of the proposed BP-Modified Local Loss algorithm is convincing.
- Experimental results indicate that the proposed method significantly improves the performance of local loss training algorithm.
- The paper is highly readable and describes the implementation details of BP-Modified Local Loss algorithm in detail.

**Weaknesses:**

- The main motivation of local training algorithms is to train large datasets or large models in resource-constrained environments. So I think authors could scale to some datasets larger than CIFAR-100, such as the ImageNet, to better illustrate the applicability of the proposed method and meet the design motivation.
- The optimization of the Loss function tends to significantly affect the training convergence performance. So I think it is necessary to provide model training time comparison results between the BP-Modified Local Loss method and other Local Loss baseline methods, such as total epochs needed for convergence.

**Questions:**

Overall, I think this is a good paper and I would raise rating if authors could address my concerns above. In addition, I suggest adding some figure descriptions if possible so that the readers could more clearly understand the authors’ intention.

---

> ### Author Response · Authors · 2024-11-26
>
> Dear Reviewer UZvZ,
>
> Thank you for reviewing the paper. We appreciate the time and effort dedicated to reviewing this draft and value your insightful comments. Following is a point-by-point response to the reviewer’s comments.
>
> 1. **motivation of local training algorithms is to train large datasets or large models in resource-constrained environments. So I think authors could scale to some datasets larger than CIFAR-100, such as the ImageNet, to better illustrate the applicability of the proposed method and meet the design motivation.**
>
>     Thank you for pointing this out. Due to the resources limitations, we have added additional experiments on the ImageNet-Tiny dataset with 10 epochs, which is usually enough for fine-tuning. The results are shown in Table 2 which is consistent with the results in CIFAR100 dataset. We found that the BP-modified local loss method can achieve better performance than the original local loss methods in most cases, especially when the original method cannot perform well. We observed that the LoKr tuning in original methods have comparable or better test accuracy on the ImageNet-Tiny dataset compared to the CIFAR100 dataset, while in BP tuning and other two model settings, the performance is much lower than the CIFAR100 dataset. We do not know what causes this phenomena and this maybe the reason why the BP-modified local loss method has lower performance than the original local loss methods in LoKr tuning case.
>
> 2. **The optimization of the Loss function tends to significantly affect the training convergence performance. So I think it is necessary to provide model training time comparison results between the BP-Modified Local Loss method and other Local Loss baseline methods, such as total epochs needed for convergence.**
>
>     you for pointing this out, we have added Table 5 to show the time overhead of the BP-modified local loss method and the original methods. It shows that the BP-modified local loss method only has a slight additional time overhead compared to the original methods.
>
>     The local loss method may not be able to converge to the same loss as the BP algorithm since the local gradient is biased. We report the average epoch that the BP-modified local loss method first achieved the same test accuracy as the final test accuracy of the original methods in CIFAR100 dataset in Table 8.
>
>     We find that the BP-modified local loss method requires much less epochs to reach the same test accuracy as the final test accuracy of the original methods, demonstrates the efficiency of the proposed method.
>
> 3. **In addition, I suggest adding some figure descriptions if possible so that the readers could more clearly understand the authors’ intention.**
>
>     Thank you for pointing this out. We have added Figure 1 on Page 3 which describes the idea of the BP-modified local loss method.
>
>     As the figure demonstrates, to modify the local gradient in batch $B$ which usually has large bias with the true gradient, the BP-modified local loss method will use additional information of local gradient and true gradient in batch $B'$.  It will compute the offset $\Delta \tilde g_{h}$ in batch $B'$ to adjust the local gradient in batch $B$. Finally, the modified local gradient will be used by the optimizers to update the model parameters.
>
>     The figure suggests the idea of the BP-modify local loss method and the lazy update technique used in the method. The bias-variance trade-off is not presented since it is haed to combine the gradient and its distribution property in one figure.

---

> ### Comment · Reviewer_UZvZ · 2024-11-27
>
> Thank you for your reponse, which resolved all of my doubts. I am willing to raise my rating and recommend to accept this paper.

---

### Official Review · Reviewer_EzNV · 2024-10-27

**Soundness:** 3
**Presentation:** 2
**Contribution:** 3
**Rating:** 5
**Confidence:** 3

**Summary:**

In this paper, authors proposed a novel training algorithm called BP-modified local loss, which aims to address non-convergence issues and poorer performance in the previous local loss methods, e.g., GIM, LoCo, and Forward-Forward algorithm. It combines backward propagation (BP) with local loss methods by modifying the local gradient using the BP gradient to balance bias and variance.

**Strengths:**

1. Strong empirical results: The BP-modified local loss shows significant improvement, e.g., up to 36% improvement in the test accuracy for Forward-Forward algorithm, 20% for LoCo and 11% for MPC.

2. The use of scaling factors $\alpha_{h}$ and $\lambda_{h}$ to manage the bias-variance balance is well motivated. The use of stochastic differential equations to model the modified training dynamic helps in understanding the benefits, as well as the limitations of the method.

**Weaknesses:**

1. Generalisability is one of the major concerns, as the scope of the experiments is a bit limited. The results are primarily conducted on CIFAR-100, a relatively small dataset. While the improvements are evident, a more robust evaluation that includes larger datasets like ImageNet, as well as tasks other than image classification, would make the proposed method more convincingly.

2. The authors leveraged the Ornstein-Uhlenbck process to model the bias-variance dynamic, however, the OU process is a simplified linear model, and deep neural network training is inherently non-linear and non-convex. The validity of the approximation that discussed in the paper would require some empirical evidence to support, as well as some discussions in more depth. For example, deep neural networks exhibit different training dynamics across layers, with shallow and deep layers experiencing different gradient behaviours. Can authors provide how gradients evolve for different layers and test if the OU process holds similarly for both shallow and deep layers.

**Questions:**

1. The method introduced several new hyper-parameters, such as batch size $B^{'}$ for the BP gradient, scaling factors $\lambda_{h}$ and $\alpha_{h}$, as well as the lazy update period $K$. I wonder how sensitive is the performance to these hyper-parameters, can authors report sensitivity analyses for these hyper-parameters? And what guidance or insight can authors provide on how to tune them for different models and datasets? Specifically, the optimal scaling factors, $\lambda_{h}$ and $\alpha_{h}$, are derived theoretically, but their practical implementation seems challenging.

2. Although the authors adopted some strategies like lazy update and split mini-batch, computing the offset $Δg_{h,t}$ involves additional operations. I wonder what’s the overhead in terms of runtime and memory usage as the depth of the network increases? Can authors quantify the impact of these additional operations on the overall training efficiency? Such as total training time, per-epoch time as well as GPU memory usage as the depth of the network increases?

---

> ### Author Response · Authors · 2024-11-26
>
> Dear Reviewer EzNV,
>
> Thank you for reviewing the paper. We appreciate the time and effort dedicated to reviewing this draft and value your insightful comments. Following is a point-by-point response to the reviewer’s comments.
>
> 1. **Generalisability is one of the major concerns, as the scope of the experiments is a bit limited. The results are primarily conducted on CIFAR-100, a relatively small dataset. While the improvements are evident, a more robust evaluation that includes larger datasets like ImageNet, as well as tasks other than image classification, would make the proposed method more convincing.**
>
>     Thank you for pointing this out. Due to the resources limitations, we have added additional experiments on the ImageNet-Tiny dataset with 10 epochs, which is usually enough for fine-tuning. The results are shown in Table 2 which is consistent with the results in CIFAR100 dataset. We found that the BP-modified local loss method can achieve better performance than the original local loss methods in most cases, especially when the original method cannot perform well. We observed that the LoKr tuning in original methods have comparable or better test accuracy on the ImageNet-Tiny dataset compared to the CIFAR100 dataset, while in BP tuning and other two model settings, the performance is much lower than the CIFAR100 dataset. We do not know what causes this phenomena and this maybe the reason why the BP-modified local loss method has lower performance than the original local loss methods in LoKr tuning case.
>
> 2. **The authors leveraged the Ornstein-Uhlenbck process to model the bias-variance dynamic, however, the OU process is a simplified linear model, and deep neural network training is inherently non-linear and non-convex. The validity of the approximation discussed in the paper would require some empirical evidence to support, as well as some discussions in more depth. For example, deep neural networks exhibit different training dynamics across layers, with shallow and deep layers experiencing different gradient behaviours. Can authors provide how gradients evolve for different layers and test if the OU process holds similarly for both shallow and deep layers.**
>
>     Yes, the OU process is a much simplified linear model and there is no evidence that the training process of any neural network will be similar to an OU process.
>
>     However, the OU process is just a toy example to introduce the bias-variance trade-off analysis. The result of the OU process gives us some insight into the bias-variance trade-off and the sub-optimality of the unbiased modification. These are also consistent with the one-step loss analysis in Section 4.2. The one-step loss analysis is the main analysis in our paper and it can apply to any deep neural network and task with Lipschitz continuous gradient and the variances of the local gradient and true gradient being bounded.
>
>     Thank you for pointing this out. We have revised the introduction paragraph in Section 4.1. The different training dynamics across layers is a good point and we will consider this in our future work.

---

> > ### Author Response · Authors · 2024-11-26
> >
> > 3. **The method introduced several new hyper-parameters, such as batch size $B'$ for the BP gradient, scaling factors $\lambda_h$ and $a_h$, as well as the lazy update period $K$. I wonder how sensitive is the performance to these hyper-parameters, can authors report sensitivity analyses for these hyper-parameters? And what guidance or insight can authors provide on how to tune them for different models and datasets? Specifically, the optimal scaling factors, $\lambda_h$ and $a_h$, are derived theoretically, but their practical implementation seems challenging.**
> >
> >     For the sensitivity of $B'$ and $K$, please refer to the Global Response 2.
> >
> >     Theoretically, a larger period $K$, or equivalently, a lazier update of the $\Delta \tilde g_{h}$ will lead to larger variance since the noise of the same offset will accumulate, and enhance the delayed bias, thus reducing the performance. On the other hand, as Eq. (7) suggests, a smaller $B'$ will also increase the variance. However, a smaller $K$ and larger $B'$ mean more computation.
> >
> >     We have added some numerical results on the sensitivity of the period $K$ and mini-batch size $B'$ and the results are shown in Appendix D.2. The results align with the previous discussion that larger $K$ and smaller $B'$ save some time at the expense of lower performance and vice versa.
> >
> >     Additionally, if the mini-batch size $B'$ is too small, the variance of the offset will be large and the performance might be worse, while it seems that it has more tolerance for larger $K$. If the period $K$ is too small or $B'$ is too large, the computation overhead will be large with diminishing performance improvement. Our experience suggests that $B'$ should be large enough to reduce the variance of the offset while $K$ can be relatively large such that the computation overhead is small. $B'$ being $1\%$ of $BK$ are usually good choices.
> >
> >     For the implement of the scaling factors $\lambda_h$ and $a_h$, as shown in Eq. (17)(18)(19), their optimal values only depend on the norm true gradient and local gradient $\|g_h\|,|g_T|$ ; the inner product between these two gradients $g_{h}^\top g_{T}$: and the trace of variance of these two gradients and their covariance $tr(Cov(g_h)),tr(Cov(g_T)),tr(Cov(g_h,g_T))$. These values can be easily estimated when acquiring offset $\Delta \tilde g_{h}$. Especially noted that the trace of the covariance $C_X$ of a random vector $X$ is equal to the expectation of the square norm of $X$ minus the square of the expectation of $X$, i.e. $tr(C_X)=\mathbb{E}[\|X\|_2^2]-\|\mathbb{E}[X]\|_2^2$ and $tr(Cov(g_h-g_T))=tr(Cov(g_h))+tr(Cov(g_T))-2tr(Cov(g_h,g_T))$.
> >
> >     Furthermore, Eq. (17)(18) are coupled but since Eq. (16) is convex and quadratic for both $a$ and $\lambda$, the optimal solution can be obtained by iteratively solving them for a few steps.

---

> ### Author Response · Authors · 2024-11-26
>
> 4. **Although the authors adopted some strategies like lazy update and split mini-batch, computing the offset $\Delta \tilde g_{h}$ involves additional operations. I wonder what’s the overhead in terms of runtime and memory usage as the depth of the network increases? Can authors quantify the impact of these additional operations on the overall training efficiency? Such as total training time, per-epoch time as well as GPU memory usage as the depth of the network increases?**
>
>     The additional relative computation overhead of the BP-modified local loss method will be about $\frac{(1+r)B'}{BK}$ where $r$ is the ratio of computation overhead between BP algorithm and local loss algorithm on one sample. The memory overhead will be on the scale of $O(Bh,b'T)$ where $b'$ is the batch size after splitting, and the additional memory usage is the size of trainable parameters.
>
>     Since we use the lazy update technique, the additional computation will be the acquisition of the local gradient $\tilde{g}_h$ and the true gradient $\tilde{g}_T$ in the batch $B'$ every $K$ steps. Therefore, the additional relative computation overhead is $\frac{(1+r)B'}{BK}$ where $r$ is the ratio of computation overhead between the BP algorithm and the local loss algorithm on one sample. The additional computation required for the scaling factors $\lambda_h$ and $a_h$ like $\|g_h\|_2,tr(Cov(g_h))$ mentioned in the previous response is of the size of the trainable parameters and is usually negligible.
>
>     Furthermore, since we use the mini-batch splitting technique to split $B'$ into smaller mini-batches, it will not change the total computation but will slightly increase the computation time than no split. Then the additional time overhead will be slightly larger than $\frac{(1+r)B'}{BK}$. We have added Table 5 in Appendix D.1 to show the time overhead of the BP-modified local loss method and the original methods.
>
>     As the model becomes deeper, the only parameter involved in computation that will change is $r$, the ratio of computation overhead between the BP algorithm and local loss algorithm on one sample. However, how $r$ changes with the depth of the network depends on the specific model and the design of the local loss method.
>
>     On the other hand, our method needs additional memory to store the offset $\Delta \tilde g_{h}$ which is the size of trainable parameters and is usually negligible, especially for CNN and low-rank tuning cases such as LoRA. The scaling factors are some float or double numbers which are also negligible.
>
>     As for the memory usage, the memory usage will increase linearly with batch size $B$ and it is reported that the memory usage will grow linearly with respect to horizon $h$, and these two factors are independent. When computing the offset $\Delta \tilde g_{h}$, the memory usage in getting local gradient $\tilde g_{h}$ and true gradient $\tilde{g}_{T}$ are $O(B'h),O(B'T)$ respectively. Since we use the mini-batch splitting technique to split $B'$ into smaller mini-batches with size $b'$, the memory usage will be $O(b'T)$. Further, acquiring different gradients is asynchronous so we only need to consider the maximum for different gradients, i.e. O(Bh,b'T). As long as $b'$ is small enough like $b'T\leq Bh$, the memory usage in computing gradients in batch $B'$ will be comparable to or less than computing local gradient in batch $B$.
>
>     As the depth of the model increases, $T$ will become larger, however, as long as $b'T\leq Bh$, the memory usage in computing gradients in batch $B'$ will be comparable to or less than computing local gradient in batch $B$. In the extreme case, when $T\ll Bh$, the memory overhead $O(b'T)$ will be larger than $O(Bh)$ for the original method.
>
>     For the numerical results, the ViT-b16 model and ResNet-50 model have different depths (12 and 17), and the results in Table 4 in Appendix D.1 show that the additional memory overhead is mainly the trainable parameters and negligible. There is also no significant additional computation overhead difference between the two models as Table 5 shows.
>
>     Thank you for pointing this out. We have revised the last paragraph of Section 3 to indicate the additional computation and memory overhead and Section 5 to emphasize the additional computation and memory overhead of the BP-modified local loss method.

---

> > ### Comment · Reviewer_EzNV · 2024-11-27
> >
> > Thanks for your rebuttal.
> >
> > &nbsp;
> >
> > Regarding the 'resources limitations' you've mentioned, I don't get it. According to the outputs within the 'Train.ipynb' - 'You are using a CUDA device ('NVIDIA GeForce RTX 3090') .... CUDA_VISIBLE_DEVICES: [0,1,2,3]', indicates that you have four RTX 3090 GPUs, which has 24 GB memory each. This setup should be sufficient to run ImageNet experiments for ViT-base, as well as ResNet50. Could you explain it?

---

> > > ### Author Response · Authors · 2024-11-27
> > >
> > > Thank you for your feedback.
> > >
> > > I appreciate your clarification and apologize for the earlier ambiguity regarding resource limitations. While we do have access to RTX 3090 GPUs, the primary constraint is time. Our group's policy limits each researcher to using a maximum of two GPUs simultaneously. Training on the full ImageNet dataset from scratch requires significantly more epochs than fine-tuning. For instance, the ResNet-50 paper trained for approximately 120 epochs (600k steps), and the LoCo paper extended this to 800 epochs. In contrast, our current experiments use only 30 epochs, which is sufficient for fine-tuning but far from comprehensive for full training.
> > >
> > > Due to these constraints, completing a full set of ImageNet experiments within the available timeframe was not feasible. We are actively working on additional experiments and will include more comprehensive results in our revised submission.
> > >
> > > Thank you for your understanding.

---

### Official Review · Reviewer_z11R · 2024-10-31

**Soundness:** 3
**Presentation:** 2
**Contribution:** 3
**Rating:** 6
**Confidence:** 3

**Summary:**

This paper proposes a new BP modified local loss method, which aims to adjust the local gradient by introducing an additional offset to reduce the deviation between the local gradient and the true gradient and introduce additional variance. This approach improves the performance of local loss training while maintaining memory efficiency. The theoretical analysis and experimental results of the paper show that this method can effectively improve performance on different models and tasks, while significantly reducing memory usage. This is an interesting and promising research direction with important implications for the deep learning community.

**Strengths:**

Innovation: The proposed BP modification of local loss method improves training efficiency while reducing memory usage, which is a valuable contribution.
Theoretical analysis: The paper provides a theoretical analysis based on the random modification equation, deeply explores the bias-variance trade-off, and derives the optimal scaling factor.

**Weaknesses:**

Impact of delayed gradient adjustment: The paper discusses the noise accumulation and bias adjustment problems that may be caused by delayed gradient adjustment, but does not provide a detailed analysis. It is recommended that the authors further study the impact of these factors on training dynamics and explore possible solutions.
More extensive experimental validation: Although the paper conducts experiments on multiple models and tasks, it is still recommended to conduct further experimental validation on larger models and datasets to demonstrate the universality and scalability of the method.

**Questions:**

The paper mentions hyperparameters such as offset batch size B' and sampling period K, but does not discuss the selection of these parameters in detail. It is recommended that the authors provide more analysis on how these hyperparameters affect model performance and memory usage.

---

> ### Author Response · Authors · 2024-11-26
>
> Dear Reviewer z11R,
>
> Thank you for reviewing our paper. We greatly appreciate the time and effort you have dedicated and value your insightful comments. Following is a point-by-point response to the reviewer’s comments.
>
> 1. **Impact of delayed gradient adjustment: The paper discusses the noise accumulation and bias adjustment problems that may be caused by delayed gradient adjustment, but does not provide a detailed analysis. It is recommended that the authors further study the impact of these factors on training dynamics and explore possible solutions.**
>
>     Thank you for pointing this out. The detailed analysis of the delayed gradient adjustment is much more difficult and complex since it involves multiple steps of training. The estimate in the one-step analysis may not apply to the multi-step analysis. This is beyond the scope of this paper. We will consider this in our future work.
>
>     However, we have added some numerical results on the sensitivity of the period $K$ in Appendix D.2. The results show that larger $K$ will lead to lower performance but still can achieve better performance than the original local loss methods. This observation may indicate that the bias adjustment problem is not severe in the experiment period range and that noise accumulation is the majority issue. Especially the use of momentum in the optimizer will enlarge the noise accumulation because the same accumulated noise will be considered as the accurate direction thus harder to be averaged out in momentum.
>
> 2. **More extensive experimental validation: Although the paper conducts experiments on multiple models and tasks, it is still recommended to conduct further experimental validation on larger models and datasets to demonstrate the universality and scalability of the method.**
>
>     Thank you for pointing this out. Due to the resources limitations, we have added additional experiments on the ImageNet-Tiny dataset with 10 epochs, which is usually enough for fine-tuning. The results are shown in Table 2 which is consistent with the results in CIFAR100 dataset. We found that the BP-modified local loss method can achieve better performance than the original local loss methods in most cases, especially when the original method cannot perform well. We observed that the LoKr tuning in original methods have comparable or better test accuracy on the ImageNet-Tiny dataset compared to the CIFAR100 dataset, while in BP tuning and other two model settings, the performance is much lower than the CIFAR100 dataset. We do not know what causes this phenomena and this maybe the reason why the BP-modified local loss method has lower performance than the original local loss methods in LoKr tuning case.

---

> ### Author Response · Authors · 2024-11-26
>
> 3. **The paper mentions hyperparameters such as offset batch size B' and sampling period K, but does not discuss the selection of these parameters in detail. It is recommended that the authors provide more analysis on how these hyperparameters affect model performance and memory usage.**
>
>     For the influence of period $K$ and batch size $B'$ on the performance, please refer to the Global Response 2. As discussed in Appendix C, a larger period $K$, or equivalently, a lazier update of the $\Delta \tilde{g}_{h}$ will lead to larger variance since the noise of the same offset will accumulate, and enhance the delayed bias, thus reducing the performance. On the other hand, as Eq. (7) suggests, a smaller $B'$ will also increase the variance. However, a smaller $K$ and larger $B'$ mean more computation.
>
>     We have added some numerical results on the sensitivity of the period $K$ and mini-batch size $B'$ and the results are shown in Appendix D.2. The results align with the previous discussion that larger $K$ and smaller $B'$ save some time at the expense of lower performance and vice versa.
>
>     Additionally, if the mini-batch size $B'$ is too small, the variance of the modified gradient will be large and the performance might be worse, while it seems that it has more tolerance for larger $K$. If the period $K$ is too small or $B'$ is too large, the computation overhead will be large with diminishing performance improvement. Our experience suggests that $B'$ should be large enough to reduce the variance of the modified gradient while $K$ can be relatively large such that the computation overhead is small. $B'$ being $1\%$ of $BK$ are usually good choices.
>
>     Note that there is no memory usage difference between different choices of $K$ and $B'$. Since we use the mini-batch splitting technique to split $B'$ into smaller mini-batches, the memory overhead of computing offset $\Delta \tilde g_h$ in batch $B'$ will be comparable to or less than computing local gradient in batch $B$. We do not change the smaller batch size so larger $B'$ means only more computation but not more memory usage. The additional memory usage of our method is to store the offset $\Delta \tilde{g}_{h}$ which is the size of trainable parameters, and scaling factors which are some double numbers. They are independent with the choice of $K$ and $B'$.
>
>     The theoretical analysis of the influence of period $K$ and mini-batch size $B'$ is a challenging task and beyond the scope of this paper. We will consider this in our future work.

---

### Official Review · Reviewer_Zuvr · 2024-11-03

**Soundness:** 3
**Presentation:** 3
**Contribution:** 3
**Rating:** 6
**Confidence:** 3

**Summary:**

The paper proposes a method to compute local loss based on BP which significantly improves performance of local loss based training methods with negligible memory overhead.

**Strengths:**

Significantly improved performance compared to SoTA local loss based learning methods

Results in negligible additional memory usage

Strong theocratical foundations for proposed method

**Weaknesses:**

Compare throughput with SoTA local loss-based learning methods

Misc:
Paraphrase line 31, 32, and 33 for better readibility

**Questions:**

Is there any overhead in latency or throughput for BP-modified loss compared to other local loss-based training methods?

Is the bias-variance trade-off analysis generalizable across various model architectures and tasks, or are there conditions under which it may not apply?

Can you further explain how using true gradient information doesn’t increase memory usage?

The lazy update and mini-batch splitting techniques are introduced to manage memory and computational costs. How sensitive is the method to the choice of period K and mini-batch size B'?

---

> ### Author Response · Authors · 2024-11-26
>
> Dear Reviewer Zuvr,
>
> Thank you for reviewing our paper. We appreciate the time and effort you have dedicated and value your insightful comments. Below is our point-by-point response to each of your comments:
>
> 1. **Misc: Paraphrase lines 31, 32, and 33 for better readability.**
>
>     Thank you for pointing this out. We have revised the sentence for better readability:
>
>     "This issue becomes more severe as the model becomes deeper and requires larger batch sizes, like SimCLR (Chen et al., 2020). Consequently, reducing the memory footprint of training and fine-tuning remains an active area of research (Krizhevsky et al., 2017; Rhu et al., 2016; Malladi et al., 2023)."
>
> 2. **Is there any overhead in latency or throughput for BP-modified loss compared to other local loss-based training methods?**
>
>     Yes, the additional relative computation overhead of the BP-modified local loss method will be about $\frac{(1+r)B'}{BK}$ where $r$ is the ratio of computation overhead between the BP algorithm and the local loss algorithm on one sample. The memory overhead will be on the scale of $O(Bh,b'T)$ where $b'$ is the batch size after splitting, and the additional memory usage is the size of trainable parameters.
>
>     The additional computation and memory requirement of the method is mainly on the offset $\Delta \tilde g_{h}$ on batch $B'$. Since we use the lazy update technique, the additional computation will be the acquisition of the local gradient $\tilde{g}_h$ and the true gradient $\tilde{g}_T$ in the batch $B'$ every $K$ steps. Therefore, the additional relative computation overhead is $\frac{(1+r)B'}{BK}$ where $r$ is the ratio of computation overhead between the BP algorithm and the local loss algorithm on one sample.
>
>     However, since we use the mini-batch splitting technique to split $B'$ into smaller mini-batches, it will not change the total computation but will slightly further increase the computation time. Then the additional time overhead will be slightly larger than $\frac{(1+r)B'}{BK}$. We have added Table 5 in Appendix D.1 to show the time overhead of the BP-modified local loss method and the original methods.
>
>     On the other hand, our method needs additional memory to store the offset $\Delta \tilde{g}_{h}$ which is the size of trainable parameters and is usually negligible, especially for CNN and low-rank tuning cases such as LoRA.
>
>     As for the memory usage, the memory usage will increase linearly with batch size $B$ and it is reported that the memory usage will grow linearly with respect to horizon $h$, and these two factors are independent. When computing the offset $\Delta \tilde g_{h}$, the memory usage in getting local gradient $\tilde g_{h}$ and true gradient $\tilde{g}_{T}$ are $O(B'h),O(B'T)$ respectively. Since we use the mini-batch splitting technique to split $B'$ into smaller mini-batches with size $b'$, the memory usage will be $O(b'T)$. Further, acquiring different gradients is asynchronous so we only need to consider the maximum for different gradients, i.e. $O(Bh,b'T)$. As long as $b'$ is small enough like $b'T\leq Bh$, the memory usage in computing gradients in batch $B'$ will be comparable to or less than computing local gradient in batch $B$.
>
>     Thank you for pointing this out. We have revised the last paragraph of Section 3 to indicate the additional computation and memory overhead and Section 5 to emphasize the additional computation and memory overhead of the BP-modified local loss method.
>
> 3. **Is the bias-variance trade-off analysis generalizable across various model architectures and tasks, or are there conditions under which it may not apply?**
>
>     Yes, the bias-variance trade-off analysis is generalizable across various model architectures and tasks since it is based on the stochastic modified equation Eq. (6) (7) and the one-step loss analysis in Section 4.2.
>
>     The Stochastic Modified Equation Eq. (6) was previously proposed by Li et al and Mandt et al and it is a general model that will be satisfied if the model has uniformly bounded gradient and Hessian which is satisfied by most of the modern models and tasks. Further, Eq. (7) is its natural extension in the BP-modified local loss method and it further requires the local gradient to hold the above properties.
>
>     However, the OU process in Section 4.1 is a simplified linear model to introduce the bias-variance trade-off analysis. It is just a toy example to provide insight and capture the key phenomena of bias-variance balance and the sub-optimality of the unbiased modification. There is no evidence that the training process of any deep neural network will satisfy the OU process.
>
>     Nonetheless, our main analysis is based on the one-step loss analysis in Section 4.2 where we just need the gradient of the loss function to be Lipschitz continuous, and the variance of both the true gradient and local gradient is bounded. This is a general condition that is satisfied by most of the modern models and tasks

---

> > ### Author Response · Authors · 2024-11-26
> >
> > 4. **Can you further explain how using true gradient information doesn’t increase memory usage?**
> >
> >     As mentioned in Response 2, we need additional memory to store the offset which is the size of trainable parameters and is usually negligible, especially for CNN and low-rank tuning cases such as LoRA. The scaling factors are some float or double numbers which are also negligible.
> >
> >     For the memory overhead during training, the memory usage will increase linearly with batch size $B$ and it is reported that the memory usage will grow linearly with respect to horizon $h$, and these two factors are independent. Secondly, acquiring different gradients is asynchronous so we only need to consider the maximum for different gradients, the memory usage will be $O(Bh,B'h,B'T)$. We use mini-batch splitting to split $B'$ into smaller mini-batches with size $b'$ to reduce memory usage. Then the memory usage is $O(Bh,b'T)$ ($h\leq T$), as long as $b'$ is small enough like $b'T\leq Bh$, the memory usage in computing gradients in batch $B'$ will be comparable to or less than computing local gradient in batch $B$.
> >
> > 5. **The lazy update and mini-batch splitting techniques are introduced to manage memory and computational costs. How sensitive is the method to the choice of period $K$ and mini-batch size $B'$?**
> >
> >     Please refer to the Global Response 2. We have added some numerical results on the sensitivity of the period $K$ and mini-batch size $B'$ and the results are shown in Appendix D.2. The results show that larger $K$ and smaller $B'$ save some time at the expense of lower performance and vice versa.
> >
> >     Additionally, if the mini-batch size $B'$ is too small, the variance of the offset will be large and the performance might be worse, while it seems that it has more tolerance for larger $K$. If the period $K$ is too small or $B'$ is too large, the computation overhead will be large with diminishing performance improvement. Our experience suggests that $B'$ should be large enough to reduce the variance of the offset while $K$ can be relatively large such that the computation overhead is small. $B'$ being 1% of $BK$ are usually good choices.

---

> > > ### Comment · Reviewer_Zuvr · 2024-12-02
> > >
> > > Dear Authors,
> > >
> > > Thank you for providing a detailed and thoughtful response. I have reviewed your rebuttal thoroughly and am satisfied with the explanations and revisions you have provided.

---

### Author Response · Authors · 2024-11-26
**Global Rebuttal**

Dear Reviewers,

We would like to express our sincere gratitude for your insightful comments and suggestions. We appreciate the time and effort you have dedicated to reviewing our paper. We have carefully considered each of your comments and have made the necessary revisions to improve the quality of the paper. Below is a global response to the common concerns raised by the reviewers:

1. **Modifications of the new revision (which is shown in blue color) include:**
     - We have some expressions for better readability (e.g. Lines 31-33).
    - We have revised the last two paragraphs of Section 3 that emphasize the effect and impact of the BP-modified local loss method.
    - We have added Figure 1 which describes the idea of the BP-modified local loss method.
    - We have revised Section 5 to highlight the effect of lazy update and mini-batch splitting techniques on the memory and computation overhead.
    - We have added additional experiments on the ImageNet-Tiny dataset. The results are shown in Table 2 in Section 4.2.
    - We have moved the memory and time results in Appendix D.1 to save space and added Table 5 to show the time overhead of the BP-modified local loss method and the original methods.
    - We have added some numerical results on different period $K$ and mini-batch size $B'$ in Appendix D.2 to show the influence of these hyperparameters.
    - We have reported the first epoch that the proposed method need to achieve the final test accuracy of the original methods in Appendix D.3 to show the efficiency of the proposed method.

2. **How sensitive is the method to the choice of period K and mini-batch size B', how do these hyperparameters affect model performance and memory usage.**

    **Response:** As discussed in Appendix C, a larger period $K$, or equivalently, a lazier update of the $\Delta \tilde{g}_{h}$ will lead to larger variance since the noise of the same offset will accumulate, and enhance the delayed bias, thus reducing the performance. On the other hand, as Eq. (7) suggests, a smaller $B'$ will also increase the variance. However, a smaller $K$ and larger $B'$ mean more computation.

    We have added some numerical results on the sensitivity of the period $K$ and mini-batch size $B'$ and the results are shown in Appendix D.2. The results align with the previous discussion that larger $K$ and smaller $B'$ save some time at the expense of lower performance and vice versa.

    Additionally, if the mini-batch size $B'$ is too small, the variance of the offset will be large and the performance might be worse, while it seems that it has more tolerance for larger $K$. If the period $K$ is too small or $B'$ is too large, the computation overhead will be large with diminishing performance improvement. Our experience suggests that $B'$ should be large enough to reduce the variance of the offset while $K$ can be relatively large such that the computation overhead is small. $B'$ being $1\%$ of $BK$ are usually good choices.

    The theoretical analysis of the influence of period $K$ and mini-batch size $B'$ is a challenging task and beyond the scope of this paper. We will consider this in our future work.

---

### Meta-Review · Area_Chair_aosJ · 2024-12-14

**Metareview:**

This paper proposes a new BP modified local loss method to improve local loss training. Specifically, it addresses the poor convergence and performance of existing local loss methods by modifying local gradients. This paper provides a theoretical analysis and derives optimal scaling factors. The proposed method exhibits strong empirical performance with marginal computational overhead. This paper is well-motivated and clearly written. This paper studies an important problem, the training of large models with limited memory, via local loss methods and provides strong empirical performance and interesting theoretical results. I recommend this paper for acceptance.

**Additional Comments On Reviewer Discussion:**

Reviewer UZvZ mentioned that the concerns were resolved by the authors’ responses. Especially, unlike the motivation, which is the training of large models, the actual experiment was conducted for a small dataset CIFAR100. So, the authors to address this provided additional experimental results on ImageNet-tiny in Section 4.2, demonstrating the benefit of the proposed method.

Reviewer EzNV’s original rating was “reject” since a username in the warning message in the IPython notebook file violated the double-blind review policy. But the AC informed the reviewer that this case is slightly below the policy violation bar. The reviewer changed the rating.

Reviewer UZvZ mentioned that more figures were needed for readability.  The authors added Figure 1 on Page 3.

Overall, the authors addressed many of the concerns raised by the reviewers and revised the manuscript accordingly.

---

### Decision · Program_Chairs · 2025-01-22

Accept (Poster)